# LEARNING LATENT TOPOLOGY FOR GRAPH MATCHING

## ABSTRACT

Graph matching (GM) has been traditionally modeled as a deterministic optimization problem characterized by an affinity matrix under pre-defined graph topology. Though there have been several attempts on learning more effective node-level affinity/representation for matching, they still heavily rely on the initial graph structure/topology which is typically obtained through heuristic ways (e.g. Delaunay or $k$-nearest) and will not be adjusted during the learning process to adapt to problem-specific patterns. We argue that such a mechanism for learning on the fixed topology may restrict the potential of a GM solver for specific tasks, and propose to learn latent graph topology in replacement of the fixed topology as input. To this end, we devise two types of latent graph generation procedures in a deterministic and generative fashion, respectively. Particularly, the generative procedure emphasizes the across-graph consistency and thus can be viewed as a matching-guided generative model. Our methods show superior performance over previous state-of-the-arts on public benchmarks.

## 1 INTRODUCTION

Being a long standing NP-hard problem (Loiola et al., 2007), graph matching (GM) has received persistent attention from the machine learning and optimization communities for many years. Concretely, for two graphs with $n$ nodes for each, graph matching seeks to solve[1]:

$$\max_{\mathbf{z}} \mathbf{z}^\top \mathbf{M} \mathbf{z} \qquad \text{s.t.} \quad \mathbf{Z} \in \{0,1\}^{n \times n}, \quad \mathbf{Hz} = \mathbf{1} \tag{1}$$

where the affinity matrix $\mathbf{M} \in \mathbb{R}_+^{n^2 \times n^2}$ encodes node (diagonal elements) and edge (off-diagonal) affinities/similarities and $\mathbf{z}$ is the column-wise vectorization form of the permutation matrix $\mathbf{Z}$. $\mathbf{H}$ is a selection matrix ensuring each row and column of $\mathbf{Z}$ summing to 1. $\mathbf{1}$ is a column vector filled with 1. Eq. (1) is the so-called quadratic assignment problem (QAP) (Cho et al., 2010). Maximizing Eq. (1) amounts to maximizing the sum of the similarity induced by matching vector $\mathbf{Z}$. While Eq. (1) does not encode the topology of graphs, Zhou & Torre (2016) further propose to factorize $\mathbf{M}$ to explicitly incorporate topology matrix, where a connectivity matrix $\mathbf{A} \in \{0,1\}^{n \times n}$ is used to indicate the topology of a single graph ($\mathbf{A}_{ij} = 1$ if there exists an edge between nodes $i$ and $j$; $\mathbf{A}_{ij} = 0$ otherwise). To ease the computation, Eq. (1) is typically relaxed by letting $\mathbf{z} \in [0,1]^{n^2}$ and keeping other parts of Eq. (1) intact. Traditional solvers to such relaxed problem generally fall into the categories of iterative update (Cho et al., 2010; Jiang et al., 2017) or numerical continuation (Zhou & Torre, 2016; Yu et al., 2018), where the solvers are developed under two key assumptions: 1) Affinity $\mathbf{M}$ is pre-computed with some non-negative metrics, e.g. Gaussian kernel, $L^2$-distance or Manhattan distance; 2) Graph topology is pre-defined as input either in dense (Schellewald & Schnörr, 2005) or sparse (Zhou & Torre, 2016) fashion. There have been several successful attempts towards adjusting the first assumption by leveraging the power of deep networks to learn more effective graph representation for GM (Wang et al., 2019a; Yu et al., 2020; Fey et al., 2020). However, to our best knowledge, there is little previous work questioning and addressing the problem regarding the second assumption in the context of *learning-based graph matching*[2]. For example, existing

---

[1]Without loss of generality, we discuss graph matching under the setting of equal number of nodes without outliers. The unequal case can be readily handled by introducing extra constraints or dummy nodes. Bipartite matching and graph isomorphism are subsets of this quadratic formulation (Loiola et al., 2007).

[2]There are some loosely related works (Du et al., 2019; 2020) on network alignment and link prediction without learning, which will be discussed in detail in the related works.

standard pipeline of keypoint matching in computer vision will construct initial topology by Delaunay triangulation or $k$-nearest neighbors. Then this topology will be freezed throughout the subsequent learning and matching procedures. In this sense, *the construction of graph topology is peeled from matching task as a pre-processing stage*. More examples can be found beyond the vision communities such as in social network alignment (Zhang & Tong, 2016; Heimann et al., 2018; Xiong & Yan, 2020) assuming fixed network structure for individual node matching in two networks.

We argue that freezing graph topology for matching can hinder the capacity of graph matching solvers. For a pre-defined graph topology, the linked nodes sometimes result in less meaningful interaction, especially under the message-passing mechanism in graph neural networks (Kipf & Welling, 2017). We give a schematic demonstration in Fig. 1. Though some earlier attempts (Cho & Lee, 2012; Cho et al., 2013) seek to adjust the graph topology under traditional non-deep learning setting, such procedures cannot be readily integrated into end-to-end deep learning frameworks due to undifferentiable nature. Building upon the hypothesis that there exists some latent topology better than heuristically created one for GM, our aim is to learn it (or its distribution) for GM. Indeed, jointly solving matching and graph topology learning can be intimidating due to the combinatorial nature, which calls for more advanced approaches.

In this paper, we propose an end-to-end framework to jointly learn the latent graph topology and perform GM, termed as deep latent graph matching (**DLGM**). We leverage the power of graph generative model to automatically produce graph topology from given features and their geometric relations, under specific locality prior. Different from generative learning on singleton graphs (Kipf & Welling, 2016; Bojchevski et al., 2018), our graph generative learning is performed in a pairwise fashion, leading to a novel matching-guided generative paradigm. **The source code will be made publicly available.**

**Contributions:** 1) We explore a new direction for more flexible GM by actively learning latent topology, in contrast to previous works using fixed topology as input; 2) Under this setting, we propose a deterministic optimization approach to learn graph topology for matching; 3) We further present a generative way to produce latent topology under a probabilistic interpretation by Expectation-Maximization. This framework can also adapt to other problems where graph topology is the latent structure to infer; 4) Our method achieves state-of-the-art performance on public benchmarks.

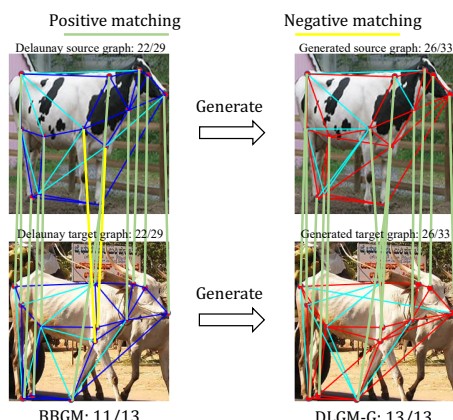

Figure 1: Matching of BBGM (Rolínek et al., 2020) 11/13 with Delaunay triangulation and our DLGM-G 13/13 using generated graph (Pascal VOC). DLGM-G generates graph with 4 more edges than Delaunay (33 vs 29) for both source and target. But with 4 more common edges across source and target than Delaunay triangulation (26 vs. 22), it leads to better accuracy. Blue and red edges denote common edges in Delaunay and learned graph pairs.

## 2    RELATED WORKS

In this section, we first discuss existing works for graph topology and matching updating whose motivation is a bit similar to ours while the technique is largely different. Then we discuss relevant works in learning graph matching and generative graph models from the technical perspective.

**Topology updating and matching.**    There are a few works for joint graph topology updating and matching, in the context of network alignment. Specifically, given two initial networks for matching, Du et al. (2019) show how to alternatively perform link prediction within each network and node matching across networks based on the observation that these two tasks can benefit to each other. In their extension (Du et al., 2020), a skip-gram embedding framework is further established under the same problem setting. In fact, these works involve a random-walk based node embedding updating and classification based link prediction modules and the whole algorithm runs in a one-shot optimization fashion. There is neither explicit training dataset nor trained matching model (except

for the link classifier), which bears less flavor of machine learning. In contrast, our method involves training an explicit model for topology recovery and matching solving. Specifically, our deterministic technique (see Sec. 3.4.1) solves graph topology and matching in one-shot, while the proposed generative method alternatively estimates the topology and matching (see Sec. 3.4.2). Our approach allows to fully leverage multiple training samples in many applications like computer vision to boost the performance on test set. Moreover, the combinatorial nature of the matching problem is not addressed in (Du et al., 2019; 2020), and they adopt a greedy selection strategy instead. While we develop a principled combinatorial learning approach to this challenge. Also their methods rely on a considerable amount of seed matchings, yet this paper directly learns the latent topology from scratch which is more challenging and seldom studied.

**Learning of graph matching.** Early non-deep learning-based methods seek to learn effective metric (e.g. weighted Euclid distance) for node and edge features or affinity kernel (e.g. Gaussian kernel) in a parametric fashion (Caetano et al., 2009; Cho et al., 2013). Recent deep graph matching methods have shown how to extracte more dedicated feature representation. The work (Zanfir & Sminchisescu, 2018) adopts VGG16 (Simonyan & Zisserman, 2014) as the backbone for feature extraction on images. Other efforts have been witnessed in developing more advanced pipelines, where graph embedding (Wang et al., 2019a; Yu et al., 2020; Fey et al., 2020) and geometric learning (Zhang & Lee, 2019; Fey et al., 2020) are involved. Rolínek et al. (2020) study the way of incorporating traditional non-differentiable combinatorial solvers, by introducing a differentiatiable blackbox GM solver (Pogancic et al., 2020). Recent works in tackling combinatorial problem with deep learning (Huang et al., 2019; Kool & Welling, 2018) also inspire developing combinatorial deep solvers, for GM problems formulated by both Koopmans-Beckmann's QAP (Nowak et al., 2018; Wang et al., 2019a) and Lawler's QAP (Wang et al., 2019b). Specifically, Wang et al. (2019a) devise a permutation loss for supervised learning, with an improvement in Yu et al. (2020) via Hungarian attention. Wang et al. (2019b) solve the most general Lawler's QAP with graph embedding technique.

**Generative graph model.** Early generative models for graph can date back to (Erdos & Renyi, 1959), in which edges are generated with fixed probability. Recently, Kipf & Welling (2016) present a graph generative model by re-parameterizing the edge probability from Gaussian noise. Johnson (2017) propose to generate graph in an incremental fashion, and in each iteration a portion of the graph is produced. Gómez-Bombarelli et al. (2018) utilized recurrent neural network to generate graph from a sequence of molecule representation. Adversarial graph generation is considered in (Pan et al., 2018; Wang et al., 2018; Bojchevski et al., 2018). Specifically, Wang et al. (2018); Bojchevski et al. (2018) seek to unify graph generative model and generative adversarial networks. In parallel, reinforcement learning has been adopted to generate discrete graphs (De Cao & Kipf, 2018).

## 3 LEARNING LATENT TOPOLOGY FOR GM

In this section, we describe details of the proposed framework with two specific algorithms derived from *deterministic* and *generative* perspectives, respectively. Both algorithms are motivated by the *hypothesis* that there exists some latent topology more suitable for matching rather than a fixed one. Note the proposed deterministic algorithm performs a standard forward-backward pass to jointly learn the topology and matching, while our generative algorithm consists of an alternative optimization procedure between estimating latent topology and learning matching under an Expectation-Maximization (EM) interpretation. In general, the generative algorithm assumes that a latent topology is sampled from a *latent distribution*, where the expected matching accuracy sufficing this distribution is maximized. Therefore, we expect to learn a topology generator sufficing such distribution. We reformulate GM into Bayesian fashion for consistent discussion in Sec. 3.1, detail deterministic/generative latent module in Sec. 3.2 and discuss the loss functions from a probabilistic perspective in Sec. 3.3. We finally elaborate on the holistic framework and the optimization procedure for both algorithms (deterministic and generative) in Sec. 3.4.

### 3.1 PROBLEM DEFINITION AND BACKGROUND

GM problem can be viewed as a Bayesian variant of Eq. (1). In general, let $\mathcal{G}^{(s)}$ and $\mathcal{G}^{(t)}$ represent the initial source and target graphs for matching, respectively. We represent graph as $\mathcal{G} := \{\mathbf{X}, \mathbf{E}, \mathbf{A}\}$, where $\mathbf{X} \in \mathbb{R}^{n \times d_1}$ is the representation of $n$ nodes with dimension $d_1$. $\mathbf{E} \in \mathbb{R}^{m \times d_2}$ are features of

$m$ edges and $\mathbf{A} \in \{0, 1\}^{n \times n}$ is initial connectivity (i.e. topology) matrix by heuristics e.g. Delaunay triangulation. For notational brevity, we assume $d_1$ and $d_2$ keep intact after updating the features across each convolutional layers of GNN (i.e., feature dimensions of both nodes and edges will not change after each layer's update). Denote the matching $\mathbf{Z} \in \{0, 1\}^{n \times n}$ between two graphs, where $\mathbf{Z}_{ij} = 1$ indicates a correspondence exists between node $i$ in $\mathcal{G}^{(s)}$ and node $j$ in $\mathcal{G}^{(t)}$, and $\mathbf{Z}_{ij} = 0$ otherwise. Given training samples $\{\mathbf{Z}_k, \mathcal{G}_k^{(s)}, \mathcal{G}_k^{(t)}\}$ with $k = 1, 2, ..., N$, the objective of learning-based GM aims to maximize the likelihood:

$$\max_\theta \prod_k P_\theta \left( \mathbf{Z}_k | \mathcal{G}_k^{(s)}, \mathcal{G}_k^{(t)} \right) \tag{2}$$

where $\theta$ denotes model parameters. $P_\theta(\cdot)$ measures the probability of matching $\mathbf{Z}_k$ given the $k$-th pair, and is instantiated via a network parameterized by $\theta$.

Being a generic module for producing latent topology, *our method can be flexibly integrated into existing deep GM frameworks*. We build up our method based on state-of-the-art (Rolínek et al., 2020), which utilizes SplineCNN (Fey et al., 2018) for node/edge representation learning. SplineCNN is a specific graph neural networks which updates a node representation via a weighted summation of its neighbors. The update rule at node $i$ of a standard SplineCNN reads:

$$(\mathbf{x} * \mathbf{g})(i) = \frac{1}{|\mathcal{N}(i)|} \sum_{l=1}^{d_1} \sum_{j \in \mathcal{N}(i)} x_l(j) \cdot g_l(\mathbf{e}(i, j)) \tag{3}$$

where $x_l(j)$ performs the convolution on node $j$ and outputs a $d_1$-dimensional feature. $g_l(\cdot)$ delivers the message weight given the edge feature $\mathbf{e}(i, j)$. $\mathcal{N}(i)$ refers to $i$'s neighboring nodes Summation over neighbors follows the topology $\mathbf{A}$. Since our algorithm learns to generate topology, we need to explicitly express Eq. (3) in a differentiable way w.r.t. $\mathbf{A}$. To this end, we rewrite Eq. (3) as:

$$(\mathbf{x} * \mathbf{g} | \mathbf{A}) = (\hat{\mathbf{A}} \circ \mathbf{G}) \hat{\mathbf{X}} \tag{4}$$

where $\hat{\mathbf{A}}$ is the normalized connectivity with each row normalized by the degree $|\mathcal{N}(i)|$ (see Eq. (3)) of the corresponding node $i$. $\mathbf{G}$ and $\hat{\mathbf{X}}$ correspond to outputs of $g_l(\cdot)$ and $x_l(\cdot)$ operators, respectively. $(\cdot \circ \cdot)$ is the Hadamard product. With Eq. (4), we thus can perform back-propagation on connectivity/topology $\mathbf{A}$. See more details in Appendix A.2.

### 3.2 LATENT TOPOLOGY LEARNING

Existing learning-based graph matching algorithms consider $\mathbf{A}$ to be fixed throughout the computation without questioning if the input topology is optimal or not. This can be problematic since input graph construction is heuristic, and it never takes into account how suitable it is for the subsequent GM task. In our framework, instead of utilizing a fixed pre-defined topology, we consider to produce latent topology under two settings: 1) a deterministic and 2) a generative way. The former is often more efficient while the latter method can be more accurate at the cost of exploring more latent topology. Note both methods produce *discrete topology* to verify our hypothesis about the existence of more suitable discrete latent topology for GM problem. The followings describe two deep structures.

**Deterministic learning**: Given input features $\mathbf{X}$ and initial topology $\mathbf{A}$, the deterministic way of generating latent topology $\underline{\mathbf{A}} \in \{0, 1\}^{n \times n}$ is[3]:

$$\underline{\mathbf{A}}_{ij} = \text{Rounding}(\text{sigmoid}(\mathbf{y}_i^\top \mathbf{W} \mathbf{y}_j)) \quad \text{with} \quad \mathbf{Y} = \text{GCN}(\mathbf{X}, \mathbf{A}) \tag{5}$$

where $\text{GCN}(\cdot)$ is the graph convolutional networks (GCN) (Kipf & Welling, 2017) and $\mathbf{y}_i$ corresponds to the feature of node $i$ in feature map $\mathbf{Y}$. $\mathbf{W}$ is the learnable parameter matrix. Note function $\text{Rounding}(\cdot)$ is undifferentiable, and will be discussed in Sec. 3.4.1.

**Generative learning**: We reparameterize the representation as:

$$P(\mathbf{y}_i | \mathbf{X}, \mathbf{A}) = \mathcal{N}(\mathbf{y}_i | \boldsymbol{\mu}_i, \text{diag}(\boldsymbol{\sigma}^2)) \tag{6}$$

---

[3]We consider the case when only node feature $\mathbf{E}$ and topology $\mathbf{A}$ are necessary. Edge feature $\mathbf{E}$ can be readily integrated as another input.

with $\boldsymbol{\mu} = \text{GCN}_{\boldsymbol{\mu}}(\mathbf{X}, \mathbf{A})$ and $\boldsymbol{\sigma} = \text{GCN}_{\boldsymbol{\sigma}}(\mathbf{X}, \mathbf{A})$ are two GCNs producing mean and covariance. It is equivalent to sampling a random vector from i.i.d. uniform distribution $\mathbf{s} \sim \mathcal{U}(\mathbf{0}, \mathbf{1})$, then applying $\mathbf{y} = \boldsymbol{\mu} + \mathbf{s} \cdot \boldsymbol{\sigma}$, where $(\cdot)$ is element-wise product.

Similar as Eq. (5) by introducing learnable parameter $\mathbf{W}$, the generative latent topology is sampled following i.i.d. distribution over each edge $(i, j)$:

$$P(\underline{\mathbf{A}}|\mathbf{Y}) = \prod_i \prod_j P(\underline{\mathbf{A}}_{ij}|\mathbf{y}_i, \mathbf{y}_j) \quad \text{with} \quad P(\underline{\mathbf{A}}_{ij} = 1|\mathbf{y}_i, \mathbf{y}_j) = \text{sigmoid}(\mathbf{y}_i^\top \mathbf{W} \mathbf{y}_j) \tag{7}$$

Since sigmoid$(\cdot)$ maps any input into $(0, 1)$, Eq. (7) can be interpreted as the probability of sampling edge $(i, j)$. As the sampling procedure is undifferentiable, we apply Gumbel-softmax trick (Jang et al., 2017) as another reparameterization procedure. As such, a latent graph topology $\underline{\mathbf{A}}$ can be sampled fully from distribution $P(\underline{\mathbf{A}})$ and the procedure becomes differentiable.

## 3.3 LOSS FUNCTIONS

In this section, we explain three loss functions and the behind motivation: *matching loss*, *locality loss* and *consistency loss*. The corresponding probabilistic interpretation of each loss function can be found in Sec. 3.4.2. These functions are selectively activated in DLGM-D and DLGM-G (see Sec. 3.4). In DLGM-G, different loss functions are activated in inference and learning steps.

**i) Matching loss**. This common term measures how the predicted matching $\hat{\mathbf{Z}}$ diverges from ground-truth $\mathbf{Z}$. Following Rolínek et al. (2020), we adopt Hamming distance on node-wise matching:

$$\mathcal{L}_M = \text{Hamming}(\hat{\mathbf{Z}}, \mathbf{Z}) \tag{8}$$

**ii) Locality loss**. This loss is devised to account for the general prior that the produced/learnt graph topology should advocate local connection rather than distant one, since two nodes may have less meaningful interaction once they are too distant from each other. In this sense, locality loss serves as a ***prior*** or regularizer in GM. As shown in multiple GM methods (Yu et al., 2018; Wang et al., 2019a; Fey et al., 2020), Delaunay triangulation is an effective way to deliver good locality. Therefore in our method, the locality loss is the Hamming distance between the initial topology $\mathbf{A}$ (obtained from Delaunay) and predicted topology $\underline{\mathbf{A}}$ for both source graph and target graph:

$$\mathcal{L}_L = \text{Hamming}(\mathbf{A}^{(s)}, \underline{\mathbf{A}}^{(s)}) + \text{Hamming}(\mathbf{A}^{(t)}, \underline{\mathbf{A}}^{(t)}) \tag{9}$$

We emphasize that locality loss serves as a *prior* for latent graph. It focuses on advocating locality, but not reconstructing the initial Delaunay triangulation (as in Graph VAE (Kipf & Welling, 2016)).

**iii) Consistency loss**. One can imagine that a GM solver is likely to deliver better performance if two graphs in a training pair are similar. In particular, we anticipate the latent topology $\underline{\mathbf{A}}^{(s)}$ and $\underline{\mathbf{A}}^{(t)}$ to be isomorphic under a specific matching, since isomorphic topological structures tend to be easier to match. Driven by this consideration, we devise the consistency loss which measures the level of isomorphism between latent topology $\underline{\mathbf{A}}^{(s)}$ and $\underline{\mathbf{A}}^{(t)}$:

$$\mathcal{L}_C(\cdot|\mathbf{Z}) = |\mathbf{Z}^\top \underline{\mathbf{A}}^{(s)} \mathbf{Z} - \underline{\mathbf{A}}^{(t)}| + |\mathbf{Z}\underline{\mathbf{A}}^{(t)}\mathbf{Z}^\top - \underline{\mathbf{A}}^{(s)}| \tag{10}$$

Note $\mathbf{Z}$ does not necessarily refer to the ground-truth, but can be any predicted matching. In this sense, latent topology $\underline{\mathbf{A}}^{(s)}$ and $\underline{\mathbf{A}}^{(t)}$ can be generated jointly given the matching $\mathbf{Z}$ as guidance information. This term can also serve as a consistency prior or regularization. We given a schematic example showing the merit of introducing consistency loss in Fig. 2(b).

## 3.4 FRAMEWORK

A schematic diagram of our framework is given in Fig. 2(a) which consists of a singleton pipeline for processing a single image. It consists of three essential modules: a feature backbone ($N_B$), a latent topology module ($N_G$) and a feature refinement module ($N_R$). Specifically, module $N_G$ corresponds to Sec. 3.2 with deterministic or generative implementations. Note the geometric relation of keypoints provide some prior for generating topology $\underline{\mathbf{A}}$. We employ VGG16 (Simonyan & Zisserman, 2014)

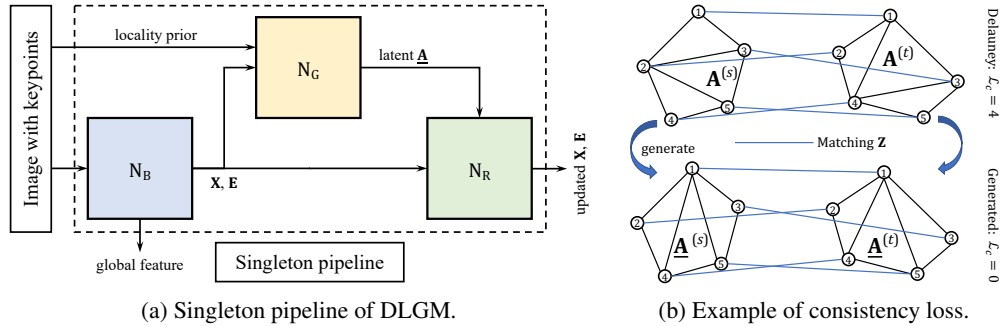

(a) Singleton pipeline of DLGM.     (b) Example of consistency loss.

Figure 2: (a) One of the two branches of our DLGM framework (see the complete version in Appendix A.1). $N_B$: VGG16 as backbone producing a global feature of input image, and initial $\mathbf{X}$ and $\mathbf{E}$; $N_G$: deterministic or generative module producing latent topology $\underline{\mathbf{A}}$; $N_R$: SplineCNN for feature refinement producing updated $\mathbf{X}$ and $\mathbf{E}$. (b) A schematic figure showing the merit of introducing consistency loss $\mathcal{L}_c$ for training. Initial topology $\mathbf{A}^{(s)}$ and $\mathbf{A}^{(t)}$ are constructed using Delaunay triangulation. Given matching $\mathbf{Z}$ as guidance, latent topology $\underline{\mathbf{A}}^{(s)}$ and $\underline{\mathbf{A}}^{(t)}$ are generated from inputs $\mathbf{A}^{(s)}$ and $\mathbf{A}^{(t)}$, respectively. Note the learned topology $\underline{\mathbf{A}}^{(s)}$ and $\underline{\mathbf{A}}^{(t)}$ are isomorphic ($\mathcal{L}_c = 0$) w.r.t. $\mathbf{Z}$ which is easier to match in test, comparing to non-isomorphic input structures ($\mathcal{L}_c = 4$).

as $N_B$ and feed the produced node feature $\mathbf{X}$ and edge feature $\mathbf{E}$ to $N_G$. $N_B$ also produces a global feature for each image. After generating the latent topology $\underline{\mathbf{A}}$, we pass over $\mathbf{X}$ and $\mathbf{E}$ together with $\underline{\mathbf{A}}$ to $N_R$ (SplineCNN (Fey et al., 2018)). The holistic pipeline handling pairwise graph inputs can be found in Fig. 4 in Appendix A.1 which consists of two copies of singleton pipeline processing source and target data (in a Siamese fashion), respectively. Then the outputs of two singleton pipelines are formulated into affinity matrix, followed by a differentiable Blackbox GM solver (Pogancic et al., 2020) with message-passing mechanism (Swoboda et al., 2017). Note once without $N_G$, the holistic pipeline with only $N_B + N_R$ is identical to the method in (Rolínek et al., 2020). Readers are referred to this strong baseline (Rolínek et al., 2020) for more mutual algorithmic details.

### 3.4.1 OPTIMIZATION WITH **DETERMINISTIC** LATENT GRAPH

We show how to optimize with deterministic latent graph module, where the topology $\underline{\mathbf{A}}$ is produced by Eq. (5). The objective of matching conditioned on the produced latent topology $\underline{\mathbf{A}}$ becomes:

$$\max \prod_k P\left(\mathbf{Z}_k | \underline{\mathbf{A}}_k^{(s)}, \underline{\mathbf{A}}_k^{(t)}, \mathcal{G}_k^{(s)}, \mathcal{G}_k^{(t)}\right) \tag{11}$$

Eq. (11) can be optimized with standard back-propagation with three loss terms activated, except for the Rounding function (see Eq. (5)), which makes the procedure undifferentiable. To address this, we use straight-through operator (Bengio et al., 2013) which performs a standard rounding during the forward pass but approximates it with the gradient of identity during the backward pass on $[0, 1]$:

$$\partial \mathrm{Rounding}(x)/\partial x = 1 \tag{12}$$

Though there exist some unbiased gradient estimators (e.g., REINFORCE (Williams, 1992)), the biased straight-through estimator proved to be more efficient and has been successfully applied in several applications (Chung et al., 2017; Campos et al., 2018). All the network modules ($N_G + N_B + N_R$) are simultaneously learned during the training. All three losses are activated in the learning procedure (see Sec. 3.3), which are applied on the predicted matching $\hat{\mathbf{Z}}$, the latent topology $\underline{\mathbf{A}}^{(s)}$ and $\underline{\mathbf{A}}^{(t)}$. We term the algorithm under this setting **DLGM-D**.

### 3.4.2 OPTIMIZATION WITH **GENERATIVE** LATENT GRAPH

See more details in Appendix A.3. In this setting, the source and target latent topology $\underline{\mathbf{A}}^{(s)}$ and $\underline{\mathbf{A}}^{(t)}$ are *sampled* according to Eq. (6) and (7). The objective becomes:

$$\max \prod_k \int_{\underline{\mathbf{A}}_k^{(s)}, \underline{\mathbf{A}}_k^{(t)}} P_\theta\left(\mathbf{Z}_k, \underline{\mathbf{A}}_k^{(s)}, \underline{\mathbf{A}}_k^{(t)} | \mathcal{G}_k^{(s)}, \mathcal{G}_k^{(t)}\right) \tag{13}$$

Unfortunately, directly optimizing Eq. (13) is difficult due to the integration over $\underline{\mathbf{A}}$ which is intractable. Instead, we maximize the evidence lower bound (ELBO) (Bishop, 2006) as follows:

$$\log P_\theta(\mathbf{Z}|\mathcal{G}^{(s)}, \mathcal{G}^{(t)}) \geq$$

$$\mathbb{E}_{Q_\phi(\underline{\mathbf{A}}^{(s)}, \underline{\mathbf{A}}^{(t)}|\mathcal{G}^{(s)}, \mathcal{G}^{(t)})} \left[ \log P_\theta(\mathbf{Z}, \underline{\mathbf{A}}^{(s)}, \underline{\mathbf{A}}^{(t)}|\mathcal{G}^{(s)}, \mathcal{G}^{(t)}) - \log Q_\phi(\underline{\mathbf{A}}^{(s)}, \underline{\mathbf{A}}^{(t)}|\mathcal{G}^{(s)}, \mathcal{G}^{(t)}) \right] \quad (14)$$

where $Q_\phi(\underline{\mathbf{A}}^{(s)}, \underline{\mathbf{A}}^{(t)}|\mathcal{G}^{(s)}, \mathcal{G}^{(t)})$ can be any joint distribution of $\underline{\mathbf{A}}^{(s)}$ and $\underline{\mathbf{A}}^{(t)}$ given the input graphs $\mathcal{G}^{(s)}$ and $\mathcal{G}^{(t)}$. Equality of Eq. (14) holds when $Q_\phi(\underline{\mathbf{A}}^{(s)}, \underline{\mathbf{A}}^{(t)}|\mathcal{G}^{(s)}, \mathcal{G}^{(t)}) = P_\theta(\underline{\mathbf{A}}^{(s)}, \underline{\mathbf{A}}^{(t)}|\mathbf{Z}, \mathcal{G}^{(s)}, \mathcal{G}^{(t)})$. For tractability, we rationally introduce the independence by assuming that we can use an identical latent topology module $Q_\phi$ (corresponding to $\mathrm{N}_G$ in Fig. 2(a)) to separately handle each input graph:

$$Q_\phi(\underline{\mathbf{A}}^{(s)}, \underline{\mathbf{A}}^{(t)}|\mathcal{G}^{(s)}, \mathcal{G}^{(t)}) = Q_\phi(\underline{\mathbf{A}}^{(s)}|\mathcal{G}^{(s)})Q_\phi(\underline{\mathbf{A}}^{(t)}|\mathcal{G}^{(t)}) \quad (15)$$

which can greatly simplify the model complexity. Then we can utilize a neural network to model $Q_\phi$ (similar to modeling $P_\theta$). The optimization of Eq. (14) is studied in (Neal & Hinton, 1998), known as the Expectation-Maximization (EM) algorithm. Optimization of Eq. (14) alternates between E-step and M-step. During E-step (inference), $P_\theta$ is fixed and the algorithm seeks to find an optimal $Q_\phi$ to approximate the true posterior distribution (see Appendix A.3 for explanation):

$$P_\theta(\underline{\mathbf{A}}^{(s)}, \underline{\mathbf{A}}^{(t)}|\mathbf{Z}, \mathcal{G}^{(s)}, \mathcal{G}^{(t)}) \quad (16)$$

During M-step (learning), $Q_\phi$ is instead fixed and algorithm alters to maximize the likelihood:

$$\mathbb{E}_{Q_\phi(\underline{\mathbf{A}}^{(s)}|\mathcal{G}^{(s)}), Q_\phi(\underline{\mathbf{A}}^{(t)}|\mathcal{G}^{(t)})} \left[ \log P_\theta(\mathbf{Z}, \underline{\mathbf{A}}^{(s)}, \underline{\mathbf{A}}^{(t)}|\mathcal{G}^{(s)}, \mathcal{G}^{(t)}) \right] \propto -\mathcal{L}_M \quad (17)$$

We give more details on the inference and learning steps as follows.

**Inference**. This step focuses on deriving posterior distribution $P_\theta(\underline{\mathbf{A}}^{(s)}, \underline{\mathbf{A}}^{(t)}|\mathbf{Z}, \mathcal{G}^{(s)}, \mathcal{G}^{(t)})$ using its approximation $Q_\phi$. To this end, we fix the parameters $\theta$ in modules $\mathrm{N}_B$ and $\mathrm{N}_R$, and only update the parameters $\phi$ in module $\mathrm{N}_G$ corresponding to $Q_\phi$. As stated in Sec. 3.2, we employ the Gumbel-softmax trick for sampling discrete $\underline{\mathbf{A}}$ (Jang et al., 2017). To this end, we can formulate a 2D vector $\mathbf{a}_{ij} = [P(\underline{\mathbf{A}}_{ij} = 1), 1 - P(\underline{\mathbf{A}}_{ij} = 1)]^\top$. Then the sampling becomes:

$$\mathrm{softmax}\left(\log(\mathbf{a}_{ij}) + \mathbf{h}_{ij}; \tau\right) \quad (18)$$

where $\mathbf{h}_{ij}$ is a random 2D vector from Gumbel distribution, and $\tau$ is a small temperature parameter. We further impose prior on latent topology $\underline{\mathbf{A}}$ given $\mathbf{A}$ through *locality loss*:

$$\log \prod_{i,j} P(\underline{\mathbf{A}}_{ij}|\mathbf{A}_{ij}) \propto -\mathcal{L}_L(\underline{\mathbf{A}}, \mathbf{A}) \quad (19)$$

which is to preserve the locality in initial topology $\mathbf{A}$. It should also be noted that $\mathbf{Z}$ is the *predicted* matching from current $P_\theta$, as $Q_\phi$ is an approximation. Besides, we also anticipate two generated topology $\underline{\mathbf{A}}^{(s)}$ and $\underline{\mathbf{A}}^{(t)}$ from a graph pair should be similar (isomorphic) given matching $\mathbf{Z}$:

$$\log P\left(\underline{\mathbf{A}}^{(s)}, \underline{\mathbf{A}}^{(t)}|\mathbf{Z}\right) \propto -\mathcal{L}_C\left(\underline{\mathbf{A}}^{(s)}, \underline{\mathbf{A}}^{(t)}|\mathbf{Z}\right) \quad (20)$$

In summary, we activate *locality loss* and *consistency loss* during the inference step, where the latter loss is conditioned with the predicted matching rather than the ground-truth. Note that the inference step involves twice re-parameterization tricks corresponding to Eq. (6) and (18), respectively. While the first generates the continuous topology distribution under edge independence assumption, the second performs discrete sampling sufficing the generated topology distribution.

**Learning**. This step optimizes $P_\theta$ by fixing $Q_\phi$. We sample discrete graph topologies $\underline{\mathbf{A}}$s completely from the probability of edge $P(\underline{\mathbf{A}}_{ij} = 1)$. Once latent topology $\underline{\mathbf{A}}$s are sampled, we feed them to module $\mathrm{N}_R$ together with the node-level features from $\mathrm{N}_B$. Only $\mathrm{N}_B$ and $\mathrm{N}_R$ are updated in this step, and only *matching loss* $\mathcal{L}_M$ is activated.

**Remark**. Note for each pair of graphs in training, we use an identical random vector $\mathbf{s}$ for generating both graphs' topology (see Eq. (6)). We pretrain the network $P_\theta$ before alternativly training $P_\theta$ and $Q_\phi$. During pretraining, we activate $\mathrm{N}_B + \mathrm{N}_R$ modules and $\mathcal{L}_M$ loss during pretraining, and feed the network the topology obtained from Delaunay as the latent topology. After pretraining, the optimization will switch between inference and learning steps until convergence. We term the setting of generative latent graph matching as **DLGM-G** and summarize it in Alg. 1.

---

Algorithm 1: Deep latent graph matching with generative latent graph (DLGM-G)

---

1: **Input:** $\mathcal{G}^s$, $\mathcal{G}^t$ and ground-truth $\mathbf{Z}$; **Output:** matching $\hat{\mathbf{Z}}$;
2: Pretrain $P_\theta$ using Eq. (11), given Delaunay as input topology;
3: **while** not converge **do**
4:     *# Inference (E-step):*
5:     Obtain predicted matching $\hat{\mathbf{Z}}$ using fixed $P_\theta$;
6:     Update $Q_\phi$ (i.e. $\text{N}_G$) with loss $\mathcal{L}_L + \mathcal{L}_C(\cdot|\hat{Z})$ according to Eq. (16);
7:     *# Learning (M-step):*
8:     Obtain predicted graph topology $\underline{\mathbf{A}}^{(s)}$ and $\underline{\mathbf{A}}^{(t)}$ using $Q_\phi$;
9:     Update $P_\theta$ (i.e. $\text{N}_B$ and $\text{N}_R$) with loss $\mathcal{L}_M$ given $\underline{\mathbf{A}}^{(s)}$ and $\underline{\mathbf{A}}^{(t)}$ according to Eq. (17);
10: **end while**
11: Predict topology and the matching $\hat{\mathbf{Z}}$ with whole network activated (i.e. $\text{N}_G + \text{N}_B + \text{N}_R$);

---

Table 1: Accuracy (%) on Pascal VOC (best in bold). Only inlier keypoints are considered.

| method | aero | bike | bird | boat | bottle | bus | car | cat | chair | cow | table | dog | horse | mbike | person | plant | sheep | sofa | train | tv | Ave |
|---|---|---|---|---|---|---|---|---|---|---|---|---|---|---|---|---|---|---|---|---|---|
| GMN | 31.1 | 46.2 | 58.2 | 45.9 | 70.6 | 76.4 | 61.2 | 61.7 | 35.5 | 53.7 | 58.9 | 57.5 | 56.9 | 49.3 | 34.1 | 77.5 | 57.1 | 53.6 | 83.2 | 88.6 | 57.9 |
| GAT-H | 47.2 | 61.6 | 63.2 | 53.3 | 79.7 | 70.1 | 65.3 | 70.5 | 38.4 | 64.7 | 62.9 | 65.1 | 66.2 | 62.5 | 41.1 | 78.8 | 67.1 | 61.6 | 81.4 | 91.0 | 64.6 |
| PCA | 40.9 | 55.0 | 65.8 | 47.9 | 76.9 | 77.9 | 63.5 | 67.4 | 33.7 | 65.5 | 63.6 | 61.3 | 68.9 | 62.8 | 44.9 | 77.5 | 67.4 | 57.5 | 86.7 | 90.9 | 63.8 |
| CIE$_1$-H | 51.2 | 69.2 | 70.1 | 55.0 | 82.8 | 72.8 | 69.0 | 74.2 | 39.6 | 68.8 | 71.8 | 66.0 | 74.4 | 68.8 | 44.8 | 85.2 | 69.9 | 65.4 | 85.2 | 92.4 | 68.9 |
| DGMC | 50.4 | 67.6 | 70.7 | 70.5 | 87.2 | 85.2 | 82.5 | 74.3 | 46.2 | 69.4 | 69.9 | 73.9 | 73.8 | 65.4 | 51.6 | 98.0 | 73.2 | 69.6 | 94.3 | 89.6 | 73.2 |
| BBGM | 61.5 | 75.0 | 78.1 | 80.0 | 87.4 | 93.0 | 89.1 | 80.2 | 58.1 | 77.6 | 76.5 | 79.3 | 78.6 | 78.8 | 66.7 | 97.4 | 76.4 | 77.5 | **97.7** | 94.4 | 80.1 |
| DLGM-D (ours) | 60.8 | 76.0 | 77.5 | 79.6 | **88.0** | **95.0** | **90.4** | 81.6 | 67.3 | 82.4 | **94.1** | 79.6 | 81.2 | 80.5 | 68.9 | **98.6** | 77.1 | 87.5 | 97.0 | 95.3 | 82.9 |
| DLGM-G (ours) | **64.7** | **78.1** | **78.4** | **81.0** | 87.2 | 94.6 | 89.7 | **82.5** | **68.5** | **83.0** | 93.9 | **82.3** | **82.8** | **82.7** | **69.6** | **98.6** | 78.9 | **88.9** | 97.4 | **96.7** | **83.8** |

Table 2: F1-score (%) on Pascal VOC. Experiment are performed on a pair of images where both inlier and outlier keypoints are considered. BBGM-max is a setting in Rolínek et al. (2020).

| method | aero | bike | bird | boat | bottle | bus | car | cat | chair | cow | table | dog | horse | mbike | person | plant | sheep | sofa | train | tv | Ave |
|---|---|---|---|---|---|---|---|---|---|---|---|---|---|---|---|---|---|---|---|---|---|
| BBGM-max | 35.5 | 68.6 | 46.7 | 36.1 | 85.4 | 58.1 | 25.6 | 51.7 | 27.3 | 51.0 | 46.0 | 46.7 | 48.9 | 58.9 | 29.6 | 93.6 | 42.6 | 35.3 | 70.7 | 79.5 | 51.9 |
| BBGM | 42.7 | 70.9 | 57.5 | 46.6 | 85.8 | 64.1 | 51.0 | 63.8 | 42.4 | 63.7 | 47.9 | 61.5 | 63.4 | 69.0 | 46.1 | 94.2 | 57.4 | 39.0 | **78.0** | 82.7 | 61.4 |
| DLGM-D (ours) | 42.5 | 71.8 | 57.8 | 46.8 | **86.9** | 70.3 | **53.4** | 66.7 | 53.8 | 67.6 | 64.7 | 64.6 | 65.2 | 70.1 | **47.9** | 95.5 | 59.6 | 47.7 | 77.7 | 82.6 | 63.9 |
| DLGM-G (ours) | **43.8** | **72.9** | **58.5** | 47.4 | 86.4 | **71.2** | 53.1 | **66.9** | **54.6** | 67.8 | 64.9 | 65.7 | 66.9 | 70.8 | 47.4 | **96.5** | **61.4** | **48.4** | 77.5 | **83.9** | **64.8** |

## 4 EXPERIMENT

We conduct experiments on datasets including Pascal VOC with Berkeley annotation (Everingham et al., 2010; Bourdev & Malik, 2009), Willow ObjectClass (Cho et al., 2013) and SPair-71K (Min et al., 2019). We report the per-category and average performance. The objective of all experiments is to maximize the average matching accuracy. Both our **DLGM-D** and **DLGM-G** are tested.

**Peer methods**. We conduct comparison experiments against the following algorithms: 1) **GMN** (Zanfir & Sminchisescu, 2018), which is a seminal work incorporating graph matching into deep learning framework equipped with a spectral solver (Egozi et al., 2013); 2) **PCA** (Wang et al., 2019a). This method treats graph matching as feature matching problem and employs GCN (Kipf & Welling, 2017) to learn better features; 3) **CIE$_1$/GAT-H** (Yu et al., 2020). This paper develops a novel embedding and attention mechanism, where GAT-H is the version by replacing the basic embedding block with Graph Attention Networks (Veličković et al., 2018); 4) **DGMC** (Fey et al., 2020). This method devises a post-processing step by emphasizing the neighborhood similarity; 5) **BBGM** (Rolínek et al., 2020). It integrates a differentiable linear combinatorial solver (Pogancic et al., 2020) into a deep learning framework and achieves state-of-the-art performance.

**Results on Pascal VOC.** The dataset (Everingham et al., 2010; Bourdev & Malik, 2009) consists of 7,020 training images and 1,682 testing images with 20 classes in total, together with the object bounding boxing for each. Following the data preparation in (Wang et al., 2019a), each object within the bounding box is cropped and resized to $256 \times 256$. The number of nodes per graph ranges from 6 to 23. We further follow (Rolínek et al., 2020) under two evaluating metrics: 1) Accuracy: this is the standard metric evaluated on the keypoints by filtering out the outliers; 2) F1-score: this metric is evaluated without keypoint filtering, being the harmonic mean of precision and recall.

Experimental results on the two setting are shown in Tab. 1 and Tab. 2. The proposed method under either settings of DLGM-D and DLGM-G outperforms counterparts by accuracy and f1-score. DLGM-G generally outperforms DLGM-D. Discussion can be found in Appendix A.5.

Table 4: Accuracy (%) on SPair-71K compared with state-of-the-art methods (best in bold).

| method | aero | bike | bird | boat | bottle | bus | car | cat | chair | cow | dog | horse | mbike | person | plant | sheep | train | tv | Ave |
|---|---|---|---|---|---|---|---|---|---|---|---|---|---|---|---|---|---|---|---|
| DGMC | 54.8 | 44.8 | 80.3 | 70.9 | 65.5 | 90.1 | 78.5 | 66.7 | 66.4 | 73.2 | 66.2 | 66.5 | 65.7 | 59.1 | 98.7 | 68.5 | 84.9 | 98.0 | 72.2 |
| BBGM | 66.9 | 57.7 | 85.8 | 78.5 | 66.9 | 95.4 | 86.1 | 74.6 | 68.3 | 78.9 | 73.0 | 67.5 | 79.3 | 73.0 | **99.1** | 74.8 | 95.0 | **98.6** | 78.9 |
| DLGM-D (ours) | 69.8 | 64.4 | **86.8** | 79.9 | **69.8** | **96.8** | **87.3** | 77.7 | 77.5 | 83.1 | 76.7 | **69.6** | **85.1** | 75.1 | 98.7 | 76.4 | 95.8 | 97.9 | 81.3 |
| DLGM-G (ours) | **70.4** | **66.8** | 86.7 | **81.7** | 69.2 | 96.4 | 85.8 | **79.5** | **78.4** | **84.0** | **79.4** | 69.4 | 84.5 | **76.6** | **99.1** | **75.9** | 96.4 | 98.5 | **82.0** |

*Quality of generated topology*. We further show the consistency/locality curve vs epoch in Fig. 3, since both consistency and locality losses can somewhat reflect the quality of topology generation. It shows that both locality and consistency losses descend during the training. Note that the consistency loss with Delaunay triangulation (green dashed line) is far more larger than our generated ones (blue/red dashed line). This clearly supports the claim that our method generates similar (more isomorphic) typologies, as well as preserving locality.

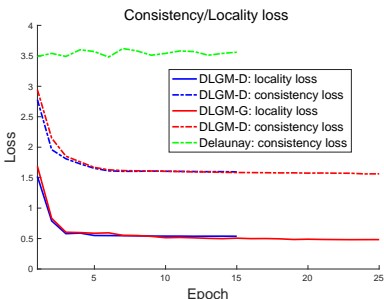

Figure 3: Consistency and locality loss (Eq. (9) and (10)) keep decrease over training showing the effectiveness for adaptive topology learning for matching.

Table 3: Accuracy (%) on Willow Object.

| Method | setting | face | mbike | car | duck | wbottle |
|---|---|---|---|---|---|---|
| GMN | Pt | 98.1 | 65.0 | 72.9 | 74.3 | 70.5 |
| | Wt | 99.3 | 71.4 | 74.3 | 82.8 | 76.7 |
| PCA | Pt | 100.0 | 69.8 | 78.6 | 82.4 | 95.1 |
| | Wt | 100.0 | 76.7 | 84.0 | 93.5 | 96.9 |
| CIE | Pt | 99.9 | 71.5 | 75.4 | 73.2 | 97.6 |
| | Wt | 100.0 | 90.0 | 82.2 | 81.2 | 97.6 |
| DGMC | Pt | 98.6 | 69.8 | 84.6 | 76.8 | 90.7 |
| | Wt | 100.0 | 98.8 | 96.5 | 93.2 | 99.9 |
| BBGM | Pt | 100.0 | 95.8 | 89.1 | 89.8 | 97.9 |
| | Wt | 100.0 | 98.9 | 95.7 | 93.1 | 99.1 |
| DLGM-D (ours) | Pt | 100.0 | 95.5 | 91.3 | 91.4 | 97.9 |
| | Wt | 100.0 | 99.4 | 95.9 | 92.8 | 99.3 |
| DLGM-G (ours) | Pt | 99.9 | 96.4 | 92.0 | 91.8 | 98.0 |
| | Wt | 100.0 | 99.3 | 96.5 | 93.7 | 99.3 |

**Results on Willow Object.** The benchmark (Cho et al., 2013) consists of 256 images in 5 categories, where two categories (car and motorbike) are subsets selected from Pascal VOC. Following the preparation protocol in Wang et al. (2019a), we crop the image within the object bounding box and resize it to $256 \times 256$. Since the dataset is relatively small, we conduct the experiment to verify the transfer ability of different methods under two settings: 1) trained on Pascal VOC and directly applied to Willow (Pt); 2) trained on Pascal VOC then finetuned on Willow (Wt). Results under the two settings are shown in Tab. 3. Since this dataset is relatively small, further improvement is difficult. It is shown both DLGM-D and DLGM-G have good transfer ability.

**Results on SPair-71K.** The dataset (Min et al., 2019) is much larger than Pascal VOC and WillowObject since it consists of 70,958 image pairs collected from Pascal VOC 2012 and Pascal 3D+ (53,340 for training, 5,384 for validation and 12,234 for testing). It improves Pascal VOC by removing ambiguous categories *sofa* and *dining table*. This dataset is considered to contain more difficult matching instances and higher annotation quality. Results are summarized in Tab. 4. Our method consistently improves the matching performance, agreeing with the results in Pascal VOC and Willow.

## 5 CONCLUSION

Graph matching involves two essential factors: the affinity model and topology. By incorporating learning paradigm for affinity/feature, the performance of matching on public datasets has significantly been improved. However, there has been little previous work exploring more effective topology for matching. In this paper, we argue that learning a more effective graph topology can significantly improve the matching, thus being essential. To this end, we propose to incorporate a latent topology module under an end-to-end deep network framework that learns to produce better graph topology. We also present the interpretation and optimization of topology module in both deterministic and generative perspectives, respectively. Experimental results show that, by learning the latent graph, the matching performance can be consistently and significantly enhanced on several public datasets.

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

## A    APPENDIX

### A.1    HOLISTIC PIPELINE

We show the holistic pipeline of our framework in Fig. 4 consisting of two "singleton pipelines" (see introduction part of Sec. 3 for more details). In general, the holistic pipeline follows the convention in a series of deep graph matching methods by utilizing an identical singleton pipeline to extract features, then exploits the produced features to perform matching (Yu et al., 2020; Wang et al., 2019a; Fey et al., 2020; Rolínek et al., 2020). Except for the topology module $N_G$, all others parts of our network are the same as those in Rolínek et al. (2020).

### A.2    SPLINECNN

SplineCNN is a method to perform graph-based representation learning via convolution operators defined based on B-splines (Fey et al., 2018). The initial input to SplineCNN is $\mathcal{G} = \{\mathbf{X}, \mathbf{E}, \mathbf{A}\}$, where $\mathbf{X} \in \mathcal{G}^{n \times d_1}$ and $\mathbf{A} \in \{0, 1\}^{n \times n}$ indicate node features and topology, respectively (same as in Sec. 3.1). $\mathbf{E} \in [0, 1]^{n \times n \times d_2}$ is so-called pseudo-coordinates and can be viewed as $n^2 \times d_2$-dimensional edge features for a fully connected graph (in case $m = n^2$, see Sec. 3.1). Let normalized edge feature $\mathbf{e}(i, j) = \mathbf{E}_{i,j,:} \in [0, 1]^{d_2}$ if a directed edge $(i, j)$ exists ($\mathbf{A}_{i,j} = 1$), and $\mathbf{0}$ otherwise ($\mathbf{A}_{i,j} = 0$). Note topology $\mathbf{A}$ fully carries the information of $\mathcal{N}(i)$ which defines the neighborhood

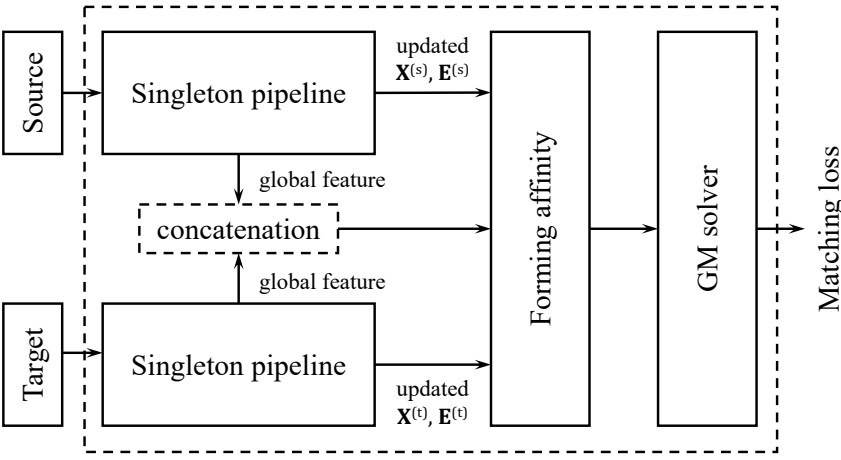

Figure 4: Holistic pipeline of DLGM consisting of two singleton pipelines.

of node $i$. During the learning, $\mathbf{X}$ and $\mathbf{E}$ will be updated while topology $\mathbf{A}$ will not. Therefore SplineCNN is a geometric graph embedding method without adjusting the latent graph topology.

B-spline is employed as basic kernel in SplineCNN, where a basis function has only support on a specific real-valued interval (Piegl & Tiller, 2012). Let $((N_{1,i}^q)_{1 \le i \le k_1}, ..., (N_{d,i}^q)_{1 \le i \le k_{d_2}})$ be $d_2$ B-spline bases with degree $q$. The kernel size is defined in $\mathbf{k} = (k_1, ..., k_{d_2})$. In SplineCNN, the continuous kernel function $g_l : [a_1, b_1] \times ... \times [a_{d_2}, b_{d_2}] \to \mathcal{G}$ is defined as:

$$g_l(\mathbf{e}) = \sum_{\mathbf{p} \in \mathcal{P}} w_{\mathbf{p},l} \cdot \mathbf{B}_{\mathbf{p}}(\mathbf{e}) \tag{21}$$

where $\mathcal{P} = (N_{1,i}^q)_i \times ... \times (N_{d,i}^q)_i$ is the B-spline bases (Piegl & Tiller, 2012) and $w_{\mathbf{p},l}$ is the trainable parameter corresponding to the $l$th node feature in $\mathbf{X}$, with $\mathbf{B}_{\mathbf{p}}$ being the product of the basis functions in $\mathbf{P}$:

$$\mathbf{B}_{\mathbf{p}} = \prod_{i=1}^{d} N_{i,p_i}^q(e_i) \tag{22}$$

where $\mathbf{e}$ is the pseudo-coordinate in $\mathbf{E}$. Then, given the kernel function $\mathbf{g} = (g_1, ..., g_{d_1})$ and the node feature $\mathbf{X} \in \mathcal{G}^{n \times d_1}$, one layer of the convolution at node $i$ in SplineCNN reads (same as Eq. (3)):

$$(\mathbf{x} * \mathbf{g})(i) = \frac{1}{|\mathcal{N}(i)|} \sum_{l=1}^{d_1} \sum_{j \in \mathcal{N}(i)} x_l(j) \cdot g_l(\mathbf{e}(i,j)) \tag{23}$$

where $x_l(j)$ indicates the convolved node feature value of node $j$ at $l$th dimension. This formulation can be tensorized into Eq. (4) with explicit topology matrix $\mathbf{A}$. In this sense, we can back-propagate the gradient of $\mathbf{A}$. Reader are referred to Fey et al. (2018) for more comprehensive understanding of this method.

### A.3 DERIVATION OF DLGM-D

We give more details of the optimization on DLGM-D in this section. This part also interprets some basic formulation conversion (e.g. from Eq. (2) to its Bayesian form). First, we assume there is no latent topology $\underline{\mathbf{A}}^{(s)}$ and $\underline{\mathbf{A}}^{(s)}$ at the current stage. In this case, the objective of GM is simply:

$$\max \prod_k P_\theta \left( \mathbf{Z}_k | \mathcal{G}_k^{(s)}, \mathcal{G}_k^{(t)} \right) \tag{24}$$

where $P_\theta$ measures the probability of a matching $\mathbf{Z}_k$ given graph pair $\mathcal{G}_k^{(s)}$ and $\mathcal{G}_k^{(t)}$. If we impose the latent topology $\underline{\mathbf{A}}^{(s)}$ and $\underline{\mathbf{A}}^{(t)}$, as well as some *distribution* over them, then Eq. (24) can be equivalently expressed as:

$$\max \prod_k P_\theta \left( \mathbf{Z}_k | \mathcal{G}_k^{(s)}, \mathcal{G}_k^{(t)} \right) = \max \prod_k \int_{\underline{\mathbf{A}}_k^{(s)}, \underline{\mathbf{A}}_k^{(t)}} P_\theta \left( \mathbf{Z}_k, \underline{\mathbf{A}}_k^{(s)}, \underline{\mathbf{A}}_k^{(t)} | \mathcal{G}_k^{(s)}, \mathcal{G}_k^{(t)} \right) \tag{25}$$

where $P_\theta \left( \mathbf{Z}_k | \mathcal{G}_k^{(s)}, \mathcal{G}_k^{(t)} \right)$ is the marginal distribution of $P_\theta \left( \mathbf{Z}_k, \underline{\mathbf{A}}_k^{(s)}, \underline{\mathbf{A}}_k^{(t)} | \mathcal{G}_k^{(s)}, \mathcal{G}_k^{(t)} \right)$ with respect to $\mathbf{Z}_k$, since $\underline{\mathbf{A}}_k^{(s)}$ and $\underline{\mathbf{A}}_k^{(t)}$ are integrated over some distribution. Herein we can impose another distribution of the topology $Q_\phi(\underline{\mathbf{A}}_k^{(s)}, \underline{\mathbf{A}}_k^{(t)} | \mathcal{G}_k^{(s)}, \mathcal{G}_k^{(t)})$ characterized by parameter $\phi$, then we have:

$$\begin{aligned}
&\log \int_{\underline{\mathbf{A}}_k^{(s)}, \underline{\mathbf{A}}_k^{(t)}} P_\theta \left( \mathbf{Z}_k, \underline{\mathbf{A}}_k^{(s)}, \underline{\mathbf{A}}_k^{(t)} | \mathcal{G}_k^{(s)}, \mathcal{G}_k^{(t)} \right) \\
&= \log \int_{\underline{\mathbf{A}}_k^{(s)}, \underline{\mathbf{A}}_k^{(t)}} P_\theta \left( \mathbf{Z}_k, \underline{\mathbf{A}}_k^{(s)}, \underline{\mathbf{A}}_k^{(t)} | \mathcal{G}_k^{(s)}, \mathcal{G}_k^{(t)} \right) \frac{Q_\phi(\underline{\mathbf{A}}_k^{(s)}, \underline{\mathbf{A}}_k^{(t)} | \mathcal{G}_k^{(s)}, \mathcal{G}_k^{(t)})}{Q_\phi(\underline{\mathbf{A}}_k^{(s)}, \underline{\mathbf{A}}_k^{(t)} | \mathcal{G}_k^{(s)}, \mathcal{G}_k^{(t)})} \\
&= \log \left( \mathbb{E}_{Q_\phi(\underline{\mathbf{A}}_k^{(s)}, \underline{\mathbf{A}}_k^{(t)} | \mathcal{G}_k^{(s)}, \mathcal{G}_k^{(t)})} \left[ \frac{P_\theta \left( \mathbf{Z}_k, \underline{\mathbf{A}}_k^{(s)}, \underline{\mathbf{A}}_k^{(t)} | \mathcal{G}_k^{(s)}, \mathcal{G}_k^{(t)} \right)}{Q_\phi(\underline{\mathbf{A}}_k^{(s)}, \underline{\mathbf{A}}_k^{(t)} | \mathcal{G}_k^{(s)}, \mathcal{G}_k^{(t)})} \right] \right) \\
&\ge \mathbb{E}_{Q_\phi(\underline{\mathbf{A}}_k^{(s)}, \underline{\mathbf{A}}_k^{(t)} | \mathcal{G}^{(s)}, \mathcal{G}^{(t)})} \left[ \log P_\theta(\mathbf{Z}, \underline{\mathbf{A}}^{(s)}, \underline{\mathbf{A}}^{(t)} | \mathcal{G}^{(s)}, \mathcal{G}^{(t)}) - \log Q_\phi(\underline{\mathbf{A}}^{(s)}, \underline{\mathbf{A}}^{(t)} | \mathcal{G}^{(s)}, \mathcal{G}^{(t)}) \right]
\end{aligned} \tag{26}$$

where the final step is derived from Jensen's inequality. Since optimizing Eq. 25 is difficult, we can alter to maximize the right hand side of inequality of Eq. (26) instead, which is the Evidence Lower Bound (ELBO) (Bishop, 2006). Since two input graphs are handled separately by two identical subroutines (see Fig. 2a), we can then impose the independence of topology $\underline{\mathbf{A}}_k^{(s)}$ and $\underline{\mathbf{A}}_k^{(t)}$: $Q_\phi(\underline{\mathbf{A}}^{(s)}, \underline{\mathbf{A}}^{(t)}|\mathcal{G}^{(s)}, \mathcal{G}^{(t)}) = Q_\phi(\underline{\mathbf{A}}^{(s)}|\mathcal{G}^{(s)})Q_\phi(\underline{\mathbf{A}}^{(t)}|\mathcal{G}^{(t)})$. In this sense, we can utilize the same parameter $\phi$ to characterize two identical neural networks (generators) for modeling $Q_\phi$.

Assuming $\theta$ is fixed, ELBO is determined by $Q_\phi$. According to Jensen's inequality, equality of Eq. (26) holds when:

$$\frac{P_\theta\left(\mathbf{Z}_k, \underline{\mathbf{A}}_k^{(s)}, \underline{\mathbf{A}}_k^{(t)}|\mathcal{G}_k^{(s)}, \mathcal{G}_k^{(t)}\right)}{Q_\phi\left(\underline{\mathbf{A}}_k^{(s)}, \underline{\mathbf{A}}_k^{(t)}|\mathcal{G}_k^{(s)}, \mathcal{G}_k^{(t)}\right)} = c \tag{27}$$

where $c \neq 0$ is a constant. We then have:

$$\int_{\underline{\mathbf{A}}_k^{(s)}, \underline{\mathbf{A}}_k^{(t)}} P_\theta\left(\mathbf{Z}_k, \underline{\mathbf{A}}_k^{(s)}, \underline{\mathbf{A}}_k^{(t)}|\mathcal{G}_k^{(s)}, \mathcal{G}_k^{(t)}\right) = c \int_{\underline{\mathbf{A}}_k^{(s)}, \underline{\mathbf{A}}_k^{(t)}} Q_\phi\left(\underline{\mathbf{A}}_k^{(s)}, \underline{\mathbf{A}}_k^{(t)}|\mathcal{G}_k^{(s)}, \mathcal{G}_k^{(t)}\right) \tag{28}$$

As $Q_\phi$ is a distribution, we have:

$$\int_{\underline{\mathbf{A}}_k^{(s)}, \underline{\mathbf{A}}_k^{(t)}} Q_\phi\left(\underline{\mathbf{A}}_k^{(s)}, \underline{\mathbf{A}}_k^{(t)}|\mathcal{G}_k^{(s)}, \mathcal{G}_k^{(t)}\right) = 1 \tag{29}$$

Therefore, we have:

$$\int_{\underline{\mathbf{A}}_k^{(s)}, \underline{\mathbf{A}}_k^{(t)}} P_\theta\left(\mathbf{Z}_k, \underline{\mathbf{A}}_k^{(s)}, \underline{\mathbf{A}}_k^{(t)}|\mathcal{G}_k^{(s)}, \mathcal{G}_k^{(t)}\right) = c \tag{30}$$

We now have:

$$\begin{aligned}
Q_\phi\left(\underline{\mathbf{A}}_k^{(s)}, \underline{\mathbf{A}}_k^{(t)}|\mathcal{G}_k^{(s)}, \mathcal{G}_k^{(t)}\right) &= \frac{P_\theta\left(\mathbf{Z}_k, \underline{\mathbf{A}}_k^{(s)}, \underline{\mathbf{A}}_k^{(t)}|\mathcal{G}_k^{(s)}, \mathcal{G}_k^{(t)}\right)}{c} \\
&= \frac{P_\theta\left(\mathbf{Z}_k, \underline{\mathbf{A}}_k^{(s)}, \underline{\mathbf{A}}_k^{(t)}|\mathcal{G}_k^{(s)}, \mathcal{G}_k^{(t)}\right)}{\int_{\underline{\mathbf{A}}_k^{(s)}, \underline{\mathbf{A}}_k^{(t)}} P_\theta\left(\mathbf{Z}_k, \underline{\mathbf{A}}_k^{(s)}, \underline{\mathbf{A}}_k^{(t)}|\mathcal{G}_k^{(s)}, \mathcal{G}_k^{(t)}\right)} \\
&= \frac{P_\theta\left(\mathbf{Z}_k, \underline{\mathbf{A}}_k^{(s)}, \underline{\mathbf{A}}_k^{(t)}|\mathcal{G}_k^{(s)}, \mathcal{G}_k^{(t)}\right)}{P_\theta\left(\mathbf{Z}_k|\mathcal{G}_k^{(s)}, \mathcal{G}_k^{(t)}\right)} \\
&= P_\theta\left(\underline{\mathbf{A}}_k^{(s)}, \underline{\mathbf{A}}_k^{(t)}|\mathbf{Z}_k, \mathcal{G}_k^{(s)}, \mathcal{G}_k^{(t)}\right)
\end{aligned} \tag{31}$$

Eq. (31) shows that, once $\theta$ is fixed, maximizing ELBO amounts to finding a distribution $Q_\phi$ approximating the posterior probability $P_\theta\left(\underline{\mathbf{A}}_k^{(s)}, \underline{\mathbf{A}}_k^{(t)}|\mathbf{Z}_k, \mathcal{G}_k^{(s)}, \mathcal{G}_k^{(t)}\right)$. This can be done by training the generator $Q_\phi$ to produce latent topology $\underline{\mathbf{A}}$ given graph pair and the matching $\mathbf{Z}$. This corresponds to the **Inference** part in Sec. 3.4.2.

## A.4 ABLATION STUDY

In this part, we evaluate the performance of DLGM-D and DLGM-G by selectively deactivating different loss functions (refer Sec. 3.3 for more details of the functions). We also conduct the test on DLGM-G using different sample size of the generator. This ablation test is conducted on Pascal VOC dataset and average accuracy is reported in Tab. 5.

We first test the performance of both settings of DLGM by selectively activate the designated loss functions. Experimental results are summarized in Tab. 5a. As matching loss $\mathcal{L}_M$ is essential for GM task, we constantly activate this loss for all settings. We see that the proposed novel losses $\mathcal{L}_C$ and $\mathcal{L}_L$ can consistently enhance the matching performance. Besides, DLGM-G indeed delivers better performance than DLGM-D under fair comparison.

We then test the impact of sample size from the generator $Q_\phi$ under DLGM-G. Experimental results are summarized in Tab. 5b. We see that along with the increasing sample size, the average accuracy ascends. The performance becomes stable when the sample size reaches over 16.

Table 5: Ablation test on Pascal VOC dataset. (a) Selectively deactivating loss functions on Pascal VOC. $\mathcal{L}_M$, $\mathcal{L}_C$ and $\mathcal{L}_L$ are selectively activated in DLGM-D and DLGM-G. "full" indicates all loss functions are activated. Average accuracy (%) is reported. (b) Average matching accuracy under different sampling sizes from the generator $Q_\phi$ with "full" DLGM-G setting.

(a) On losses

| method | Ave |
|---|---|
| DLGM-D ($\mathcal{L}_M + \mathcal{L}_C$) | 79.8 |
| DLGM-D ($\mathcal{L}_M + \mathcal{L}_L$) | 79.5 |
| DLGM-G ($\mathcal{L}_M + \mathcal{L}_C$) | 80.9 |
| DLGM-G ($\mathcal{L}_M + \mathcal{L}_L$) | 80.4 |
| DLGM-D (full) | 82.9 |
| DLGM-G (full) | 83.8 |

(b) On sample size

| #Sample | Ave |
|---|---|
| 1 | 82.5 |
| 2 | 83.2 |
| 4 | 83.2 |
| 8 | 83.5 |
| 16 | 83.8 |
| 32 | 83.7 |

| Class | |
|---|---|
| aero & bike | |
| bird & boat | |
| bottle & bus | |
| car & cat | |
| chair & cow | |
| table & dog | |
| horse & mbike | |
| person & plant | |
| sheep & sofa | |
| train & tv | |

Table 6: Matching examples of DLGM-G on 20 classes of Pascal VOC. The coloring of graphs and matchings follows the principle of Fig. 1 in the manuscript. Zoom in for better view.

## A.5 MORE VISUAL EXAMPLES AND ANALYSIS

We show more visual examples of matchings and generated topology using DLGM-G on Pascal VOC in Tab. 6 and Tab. 7, respectively. Each table follows **distinct coloring regulation** which will be detailed as follows:

- Tab. 6. For each class, the left and right images corresponds to Delaunay triangulation. The image in the middle refers to the predicted matching and generated graph topology. Cyan solid and dashed lines correspond to correct and wrong matchings, respectively. Green dashed lines are the ground-truth matchings that are missed by our model.

- Tab. 7. In this table, the leftmost and the rightmost columns correspond to original topology constructed using Delaunay triangulation. The two columns in the middle are the generated topology using our method given Delaunay triangulation as prior. Blue edges are the edges that Delaunay and generated ones have in common. Green edges corresponds to the ones that are in Delaunay but not in generated topology, while red edges are the ones that are generated but not in Delaunay.

We give some analysis for the following questions.

**In what case a different graph is generated?**

Since there are some generated graphs are identical to Delaunay, this question may naturally arise. We observe that, DLGM tends to produce an identical graph to Delaunay when objects are rarely with distortion and graphs are simple (e.g. tv, bottle and plant in Tab. 6 and last two rows in Tab. 7). However, when Delaunay is not sufficient to reveal the complex geometric relation or objects are with large distortion and feature diversity (e.g. cow and cat in Tab. 6 and person in Tab. 7), DLGM will resort to generating new topology with richer and stronger hint for graph matching. In other words, DLGM somewhat finds a way to identify if current instance pair is difficult or easy to match, and learns an adaptive strategy to handle these two cases.

**Why DLGM-G delivers better performance than DLGM-D?**

In general, DLGM-D is a deterministic gradient-based method. That is, the solution trajectory of DLGM-D almost follows the gradient direction at each iteration (with some variance from mini-batch). Though it is assured to reach a local optima, only following gradient is too greedy since generated graph is coupled with predicted matching. Besides, as the topology is discrete, the optimal continuous solution will have a large objective score gap to its nearest discrete sampled solution once the landspace of the neural network is too sharp. On the other hand, DLGM-G performs discrete sampling under feasible graph distribution at each iteration, which generally but not fully follows the gradient direction. This procedure can thus find better *discrete direction* with probability, hence better exploring the searching space. This behavior is similar to Reinforcement Learning, but with much higher efficiency. Additionally, EM framework can guarantee the convergence (Bishop, 2006).

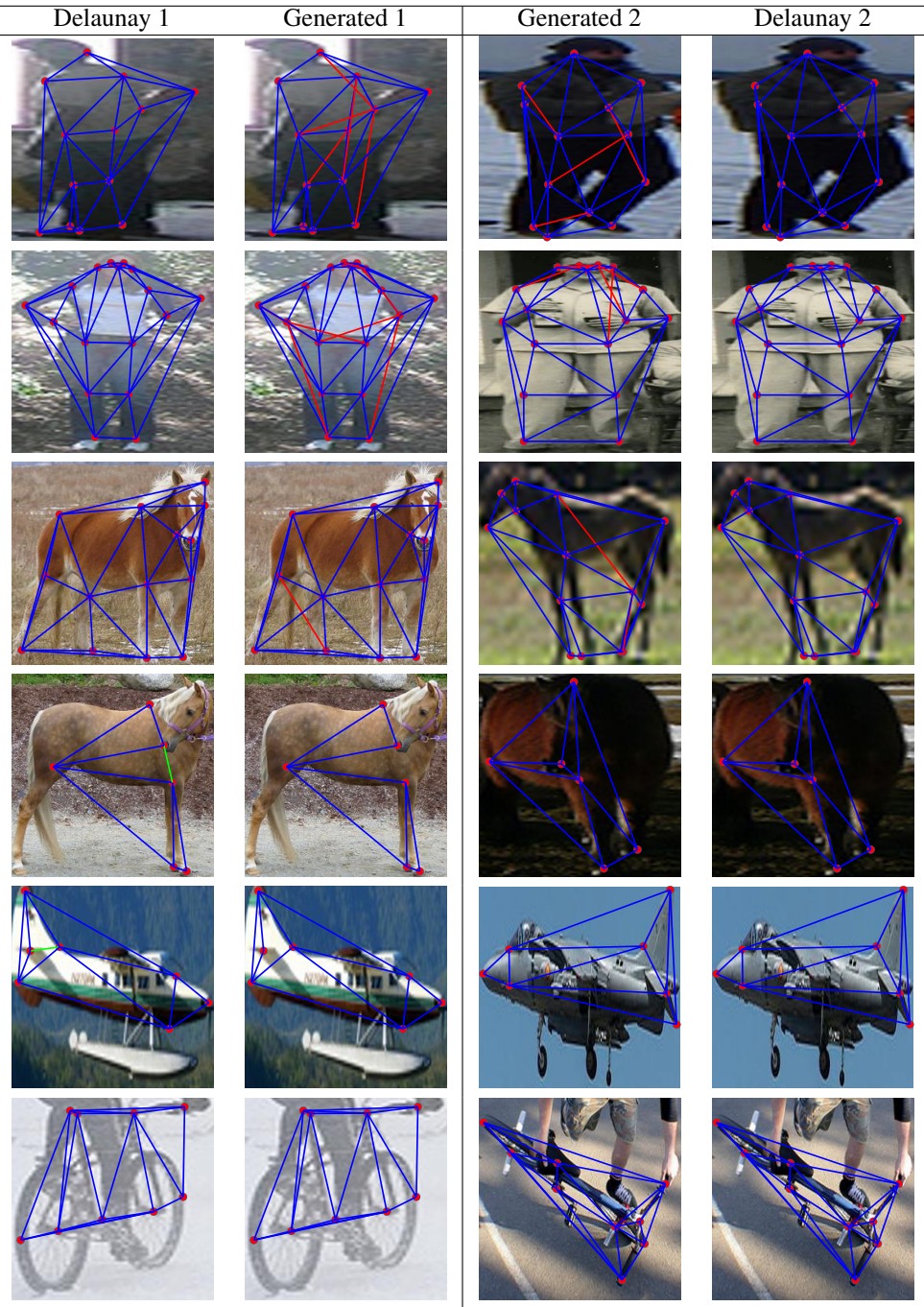

Table 7: Generated topology compared with original Delaunay triangulation in a pairwise fashion. Note the 1st and the 4th columns correspond to two input images with topology constructed by Delaunay triangulation, respectively. 2nd and 3rd columns are the generated topology given Delaunay results as prior.

