# OpenReview forum: "Learning Latent Topology for Graph Matching"
_ICLR.cc/2021/Conference — Reject_

### Official Review · AnonReviewer4 · 2020-10-28

**Rating:** 4
**Confidence:** 4

**Review:**

In this paper, the authors argue that standalone representation learning is insufficient for the Graph Matching task and propose multiple novel approaches to address this via using a latent graph topology instead of a fixed one. Empirical results demonstrate the efficacy of their approach.

I was not completely convinced by the arguments made by the authors and their motivation. Overall the paper felt a little heuristic in nature and the presentation made it difficult to follow. Here are the major aspects which were unclear to me (refer below).

1) What is the primary motivation for the authors ? Why do the authors think using a latent graph topology is in general better than using a fixed topology ? Is it always true that using a fixed topology is undesirable or only in some cases only ? Some arguments/examples from the authors which can validate this using real-world settings would have helped. Irrespective of Graph Matching problem, Social Networks can be a good application for verifying this. Fake (due to noise) and real edges in connections/friendship network would be a good indicator for the authors' intuition i.e. the latent topology should be able to filter out the fake edges.

2) With regards to eqn. 1, is the condition Hz = 1 enough to ensure that each row and column sums to 1 ?

3) Why did the authors only consider absolute/boolean assignments (0/1) in the paper rather than partial assignments which can be more flexible i.e. soft assignments (0 <= z_i <= 1) such that each row and column sums to 1 ? Is there some specific reason for this ?

4) In section 2.4.1, the authors propose to use the straight-through operator for their rounding task. Did the authors try any other approaches ? How did the bias of the straight-through operator hamper their final optimization solution and by how much ?

5) With regards to eqn. 15 wherein the authors use an independence assumption, how much did the accuracy of their estimation suffer as a result to this quantitatively ?

6) The authors also mention that their deterministic learning approach is often more efficient while the generative learning method can be more accurate at the cost of additional overhead. Do the authors have results with regards to this overhead ? How significant is it and how does it compare to time complexity of the other state-of-the-art methods ? This is even more relevant given for many of the results, the improvement is only marginal compared to other competitors.

7) Based of the results, can the authors claim that improvement in the results over other competitors is due to the efficacy of their approach and the factuality of their arguments ? Could it be due to richness/complexity of their model/learning strategy ?

8) What do the authors mean by "normalized connectivity" in section 2.1 ? Does it mean the augmented Adjacency matrix which contains self loops i.e. A_tilde = A + I ?

9) Given the authors in their approach model the adjacency matrix, thus how does the authors' approach compare against traditional graph based modeling techniques i.e. Stochastic Block Model and Erdős–Rényi model for instance ?

I would appreciate it if the authors made their code available for review/reproducing results. Additionally I felt the paper needed more analysis/discussion. Overall the presentation of the paper needs to improve significantly.

---

> ### Author Response · Authors · 2020-11-11
> **response to reviewer 4**
>
> We would also like to thank the reviewer for the effort of reviewing our paper. We response the reviewer from the following perspectives (where most of the answers can be found from the paper):
>
> 1. **Motivation.** Our idea of the paper starts from questioning if a fixed topology is suitable for a specific task (i.e. graph matching) as stated at the end of the first page. By establishing the hypothesis that there exists more suitable latent topology, we propose a method to learn it, and then verify the correctness of our hypothesis via extensive experiments. This is a process of “establishing hypothesis then verifying it”. Therefore, we didn’t “think” there must be a more suitable latent topology, instead we “conclude” clues from experiments which agree with our hypothesis.
> 2. **Fundamentals about graph matching.** $Hz=1$ does ensure a one-to-one matching when $H$ is a selection matrix and this fact has been well studied in [1,2]. In fact, as long as the model is differentiable with gradients (whenever exact gradient or gradient estimator), our method can work. In our implementation, as the black-block combinatorial solver with gradient estimator [3] is employed, the output $Z$ is indeed discrete. However,  a continuous $z_i\in [0,1]$ with exact gradient also works (e.g. [4]) and can fit into our framework. In general, our method is indifferent with the format of $Z$. A normalized connectivity is a connectivity matrix with each row normalized by the degree of the corresponding node (see Eq. (23) in Appendix A.2 with normalized value $1/|\mathcal{N}(i)|$).
> 3. **About the improvement.** In fact, we employed a completely identical state-of-the-art graph matching network structure as in [1], except for the latent topology inferring module (see $N_G$ in Fig. 1(a) and Sec 2.1). Even by only introducing this minor change, we can still achieve significant improvement (around 3%-4%) over the current state-of-the-art method [4] on challenging datasets (Pascal VOC and SPair-71K). As stated by reviewer5, these two datasets are very challenging and even an improvement around 2% is non-trival. We believe this fact is sufficiently convincing to show that the performance gain is from novel latent topology module, rather than other factors (e.g. model complexity and feature richness).
> 4. **Other gradient estimators and generative models.** Straight-through operator and the current graph generative model are adopted taking into account efficiency and simplicity. While our paper is focusing on investigating the validity of the latent topology (and its distribution), existing experimental results already can strongly support our claim and examining different combinations of basic settings is out of our scope. However, we would like to add some trials with various combinations if space and time permit.
> 5. **Independence prior.** Actually this prior is to ensure that we can use an identical neural network to model or extract features from two different input graphs. This is a common practice in almost all lines of deep graph matching methods [4,5,6], similar to Siamese networks. This procedure can greatly simplify the model complexity. In Eq. (15), we give an interpretation of such convention under a probabilistic setting. It is unnecessary and impractical to conduct tests without this prior.
> 6. **Discrete topology $\underline{\mathbf{A}}$.** Indeed, our latent topology module generates discrete topology in either deterministic or generative settings. However, our method can also easily integrate soft-assignment since it is fully differentiable. The reason we focus on the discrete case is that, since our hypothesis is about the discrete topology (and its distribution) itself, the algorithm we design should consistently align our hypothesis. Using continuous soft-assignment may possibly improve the performance, but it will also bring confusion to readers: if our claim or hypothesis is really supported by any evidence. We will consider soft-assignment in our future work.
>
> We thank the reviewer again and look forward to further discussion.
>
> [1] F. Zhou and F. Torre. Factorized graph matching. IEEE PAMI, 2016.
>
> [2] Tianshu Yu, Junchi Yan, Yilin Wang, Wei Liu, et al. Generalizing graph matching beyond quadratic assignment model. In NIPS, 2018.
>
> [3] Marin Vlastelica Pogancic, Anselm Paulus, V´ıt Musil, Georg Martius, and Michal Rol´ınek. Differentiation of black-box combinatorial solvers. In ICLR, 2020.
>
> [4] Zhen Zhang and Wee Sun Lee. Deep graphical feature learning for the feature matching problem. In ICCV, 2019.
>
> [5] Michal Rol´ınek, Paul Swoboda, Dominik Zietlow, Anselm Paulus, V´ıt Musil, and Georg Martius. Deep graph matching via blackbox differentiation of combinatorial solvers. In ECCV, 2020.
>
> [6] Tianshu Yu, Runzhong Wang, Junchi Yan, and Baoxin Li. Learning deep graph matching with channel-independent embedding and hungarian attention. In ICLR, 2020.

---

> > ### Comment · AnonReviewer4 · 2020-11-18
> > **Response**
> >
> > I would like to thank the authors for their response.
> >
> > 1) Do the authors plan to make code available for review/reproducing results ? This is pretty important.
> > 2) Do the authors plan to add results with regards to the overhead with regards to point 6 in my original review ? I would like to compare the time complexity of the techniques discussed in the paper with other state-of-the-art techniques to understand the significance of the overhead before continuing. More so since the results are marginal in many cases irrespective of the challenging datasets aspect discussed.

---

> > > ### Author Response · Authors · 2020-11-18
> > > **response to reviewer4**
> > >
> > > It is so appreciated that reviewer4 gave us response timely. For the concerns from the reviewer, our response is as follows:
> > >
> > > 1. For the current stage, the code is not ready for release since it needs more cleaning for a responsible resource sharing. But **we will release** the code with the final paper.
> > >
> > > 2. If we consider the training time of the baseline [1] to be 1x, the training time of our method under **discriminative** setting is around 1.2x-1.3x. The time cost of our method under **generative** setting is around 8x-9x with sample size 16. We didn't observe any obvious efficiency gap for the testing stage. As the reviewer said the improvement over state-of-the-art is "marginal in many cases", we wonder if the per-category performance is misleading in the tables. To clarify, the objective of our paper following [1] is to maximize the **average matching performance** (as we stated in the original paper). We didn't train for each category separately. Again, according to reviewer5, such an average improvement is non-trival.
> > >
> > > We also wonder if **other concerns** of the reviewer have been addressed, since the reviewer has raised many suggestions for us to improve the paper. It would be so appreciated if the reviewer can give us any response about these.
> > >
> > > [1] Michal Rol´ınek, Paul Swoboda, Dominik Zietlow, Anselm Paulus, V´ıt Musil, and Georg Martius. Deep graph matching via blackbox differentiation of combinatorial solvers. In ECCV, 2020.

---

> > > > ### Comment · AnonReviewer4 · 2020-11-18
> > > > **Response**
> > > >
> > > > I would like to thank the authors for their response as well as for uploading the revised submission.
> > > >
> > > > I am currently going through the authors' response to my earlier queries/concerns as well as those for the other reviewers. I will get back to them in case I have additional queries.

---

### Official Review · AnonReviewer1 · 2020-10-28
**Approach of interest but very poor presentation**

**Rating:** 4
**Confidence:** 3

**Review:**

This paper deals with graph matching, where the latent topology of the two graphs is not fixed but estimated during the learning process.

The authors propose two algorithms, namely deterministic learning and generative learning, which both achieve state-of-the-art accuracy scores on several datasets.

I think that the approach developed in this paper can be of interest but its presentation makes it difficult to follow and understand.


First, it is implicit in the first part of the paper that it deals with topologies obtained from images.
The authors should explain what the graph topology models in the image and how it is obtained in the literature. They mention algorithm names (Delaunay triangulation and k-nearest neighbors, section 1), and that the construction is heuristic (page 3), but do not provide any clue on the (formal) definition of the topology. (I recall that the paper deals with estimation of the graph topology.)


In section 2, it is proposed to estimate both the topologies of the two graphs and their matching. In my opinion, the problem is not defined in a sufficient formal way so that the reader can exactly understand the rationale behind the methods.
I also think that, at this stage, the connection with how the topology is initially estimated in the literature would both help the reader to understand the optimization problem at stake, and highlight the originality of this piece of work.

I have the following (naive) question: isn't the optimal topology the one that does not allow any edges in the graph, since it should allow the greatest flexibility in the matching of the nodes? Of course, this solution is not of interest for the applications, but I do not see why it is excluded from the investigated approach.


The authors claim that optimizing eq. (13) is intractable and thus choose to maximize a lower bound through an EM-like algorithm. However, they do not prove the relation with the initial optimization problem. As is usually the case, does it allow for local optimization of (13)?


In the real data analysis, the methods are compared through their accuracy scores and not on the estimation of the latent topology. Of course, it is irrelevant at this place since the other algorithms do not estimate it. Nevertheless, it should be interesting to have a quantitative analysis of this output of the algorithm.


For all of these reasons, I think that this paper deserves to be significantly improved before being published.


More comments and questions:

+ The reader can not read and understand Figure 1 given on page 2 without reading the paper until page 7.

+ What is the normalized connectivity (page 3)?

+ The authors mention twice in the paper that there is only little work on "exploring graph topology learning / generation for deep graph matching" (introduction and conclusion) without any references explicitly related to this sentence. Are there papers that deal with the same strategy? I believe that the article should be clearer on that point. On this topic, I also think that the structure of the paper should be revised, with the section dedicated to related works (section 4) presented earlier.

+ The qualitative analysis presented in Appendix A.5 is very interesting and should be presented in the main document.


Typos:
Page 5: which in intractable

---

> ### Author Response · Authors · 2020-11-11
> **response to reviewer 1**
>
> We would like to thank the reviewer for his/her contributing suggestions and great effort on reviewing our paper. We also appreciate that the reviewer recognized the novelty of our work. Our answers to the main concerns of the reviewer are as follows:
>
> 1. **Problem definition.** Actually the definition of learning-based graph matching under a learnable fashion (in our setting) can be found in Eq. (2), which is to estimate the matching with maximal probability given pairs of input graphs. Compared to a deterministic modeling of graph matching in Eq. (1), Eq. (2) adopts a learnable parameter $\theta$ which is more appropriate under a learning-based setting. In general, following the convention in graph neural networks [1], we denote $A\in\{0,1\}^{n\times n}$ a topology of an $n$-node graph, where $A_{ij}=1$ indicates that there is an edge between nodes $i$ and $j$, and $A_{ij}=0$ otherwise. We will clarify these in our updated version to avoid any ambiguity to the readers. A normalized connectivity is a connectivity matrix with each row normalized by the degree of the corresponding node (see Eq. (23) in Appendix A.2 with normalized value $1/|\mathcal{N}(i)|$).
> 2. **Graph without edges.** Graph without any edges will lead to bipartite matching, which is a subset of graph matching [2]. Compared to more general graph matching with edges, bipartite matching will sometimes result in the ambiguity of nodes, yielding low robustness [2]. However, it is shown in several papers that incorporating edges (topology) is helpful to the learning-based framework to fully exploit local geometrical constraints and eliminate this ambiguity to some extent [3,4]. Therefore, we mainly focus on graph matching problem with explicit topological structure.
> 3. **ELBO gaurantee.** Actually, Eq. (13) is derived from standard EM setting, except for incorporating several priors which do not have any influence on the convergence of the whole algorithm. As stated by reviewer5, the EM algorithm we utilized is standard and the contribution is where it is applied. Therefore, we omitted to rewrite the non-informative proof which can be easily found in the literature [5].
> 4. **Quality of generated topology.** Reviewer is referred to Fig. 3 which demonstrates the quality of the generated topology. We show two criteria of topology along with the epoch: 1) level of isomorphism of generated topology pairs (in terms of Eq. (10)); 2) level of how much the generated topology follows locality prior (in terms of Eq. (9)). These are two quantitative curves evaluating the quality of the generated topology. In particular, we can find that the generated topology is with higher level isomorphism compared to fixed one.
> 5. **Organization of the paper.** We will re-organize the paper to make it more friendly to readers.
> We thank the reviewer again and look forward to further discussion.
> 6. **Previous works.** Actually, we didn’t find any previous work for graph matching by estimating the latent topology as in our paper. We used “little” just to accomodate any possibility of such works. Since this can be misleading, we will correct it as “... there is no work on ...” instead. We will be happy to cite and discuss similar papers if reviewers can tell us any.
>
>
> [1] Thomas N Kipf and Max Welling. Semi-supervised classification with graph convolutional networks. In ICLR, 2017.
>
> [2] E. M. Loiola, N. M. de Abreu, P. O. Boaventura-Netto, P. Hahn, and T. Querido. A survey for the quadratic assignment problem. EJOR, pp. 657–90, 2007.
>
> [3] Michal Rol´ınek, Paul Swoboda, Dominik Zietlow, Anselm Paulus, V´ıt Musil, and Georg Martius. Deep graph matching via blackbox differentiation of combinatorial solvers. In ECCV, 2020.
>
> [4] Zhen Zhang and Wee Sun Lee. Deep graphical feature learning for the feature matching problem. In ICCV, 2019.
>
> [5] Bishop, C. M. (2006). Pattern recognition and machine learning. springer.

---

### Official Review · AnonReviewer3 · 2020-10-28
**Graph Matching, where the main contribution is to learn the connectivity matrix A, given the input features, and jointly learn the matching. To me, the use of "graph matching" is misleading**

**Rating:** 6
**Confidence:** 1

**Review:**

In this paper the authors propose a method for what they call Graph Matching, where the main contribution is to learn the connectivity matrix A, given the input features, and jointly learn the matching.

The paper is well written in general, and the contributions are clear. There are some "." and "," missing after equations.

This may be a matter of taste, but to me, graph matching (or the graph matching problem) refers to the problem of, given two graphs, finding the bijection between the set of vertices that minimices some distance. Usually, if A and B are the adjacency matrices of the two graphs, one tries to minimize $\|A-PBP^T\|_F$, over the set of permutation matrices.

Adding features to the vertices is a first extension of this problem, which I still consider a graph matching problem.
Now, if in the problem the inputs are a set of points in $\mathbb{R}^d$ and some features, and as part of the solution, one infers a possible adjacency matrix (or topology, as in the paper), then for me the term graph matching is misleading.

Unfortunately, I have no expertise in this latter problem, and therefore I just can say that the formulation seems correct, and the numerical results are promising.

---

> ### Author Response · Authors · 2020-11-11
> **response to reviewer 3**
>
> We would like to thank the reviewer for his/her contributing suggestions and great effort on reviewing our paper. We also appreciate that the reviewer recognized the novelty of our work. Our answers to the main concerns of the reviewer are as follows:
>
> **Definition of graph matching.** The graph matching formulation that the reviewer has pointed out is typically called “geometric graph matching” or “graph isomorphism” [1]. However, we consider a more generic case “quadratic assignment problem” (QAP) and graph isomorphism is in fact a subset of QAP [2]. This convention is followed by a line of works on machine learning and computer vision [3,4], where QAP is termed as "graph matching". We will clarify this.
>
> As such, we think the term "graph matching" can be an appropriate expression following other literatures.
>
> We thank the reviewer again and look forward to further discussion.
>
> [1] Babai, László. Graph isomorphism in quasipolynomial time. STOC, 2016.
>
> [2] E. M. Loiola, N. M. de Abreu, P. O. Boaventura-Netto, P. Hahn, and T. Querido. A survey for the quadratic assignment problem. EJOR, pp. 657–90, 2007.
>
> [3] F. Zhou and F. Torre. Factorized graph matching. IEEE PAMI, 2016.
>
> [4] Tianshu Yu, Junchi Yan, Yilin Wang, Wei Liu, et al. Generalizing graph matching beyond quadratic assignment model. In NIPS, 2018.

---

### Official Review · AnonReviewer2 · 2020-10-29
**Novel method for graph matching using deep networks**

**Rating:** 8
**Confidence:** 3

**Review:**

Summary: The paper discusses the problem of graph matching (GM), which is the combinatorial (NP-hard) problem of finding a similarity between graphs, and has various applications in machine learning. More specifically, the paper proposes methods to leverage the power of deep networks to come up with an end-to-end framework that jointly learns a latent graph topology and perform GM, which they term as deep latent graph matching (DLGM).

Strengths: The proposed method seems justified. The authors both explore a novel direction for GM by actively learning latent topology.  They further propose both a deterministic optimization-based approach a generative way to learn effective graph topology for matching. Regarding the empirical results of the paper, the authors report that their methods achieve state-of-the-art performance on public benchmarks, in comparison against a few peer methods (in measures of both accuracy and F1-score).
 Other than that, their claims appear to be correct, and so is the empirical methodology. Relation to prior work and differences are discussed.

Weaknesses/Comments: The paper was very difficult to follow (maybe it is due to the fact that I am not highly familiar with part of the field.  Nevertheless, I think that the organization of the paper can be improved).  I think there is a missing word in the second last sentence on page 2: For notational brevity, we assume d1 and d2 keep the same across convolutional
layers. Same dimensions maybe?

---

> ### Author Response · Authors · 2020-11-11
> **response to reviewer 2**
>
> We would like to thank the reviewer for his/her contributing suggestions and great effort on reviewing our paper. We also appreciate that the reviewer recognized the novelty of our work. And indeed $d_1$ and $d_2$ are the dimensions of node and edge features, respectively. We will re-organize our paper as suggested by the reviewer and make it more friendly to readers.

---

### Official Review · AnonReviewer5 · 2020-11-08
**Very good results on difficult benchmarks, write-up lacks clarity at times**

**Rating:** 7
**Confidence:** 4

**Review:**

The authors address the problem of discrete keypoint matching. For an input pair of images, the task is to match the unannotated (but given as part of the input) keypoints. The main contribution is identifying the bottleneck of the current SOTA algorithm: a fixed connectivity construction given by Delauney triangulation. By replacing this with an end-to-end learnable algorithm, they outperform SOTA with a decent margin.

Contribution & significance: The topic of discrete keypoint matching has had a lot of progress in the last five years and is gaining attention even from more product-oriented branches of computer vision. The benchmarks used for evaluation (PASCAL-VOC, SPair71k) are, in fact, quite difficult, and even 2-3 points of progress require non-trivial effort. The core insight of improving this particular part of the pipeline is creative and novel. In these regards, the paper certainly meets the bar set for ICLR.

Experiments are conducted thoroughly and on multiple standard datasets. I appreciate the ablation study. Minor suggestions are below.

Weaknesses:

1) **Code**  For a mostly experimental paper, it is imperative to release code (as is also common in the line of work for keypoint matching). If the code was already uploaded, I would have given the paper a higher rating.

2) **Co-generative model** The authors for some reason insist on this interpretation of their architecture (it appears in boldface in the abstract). In practical terms, it means that a) one term of the loss function doesn't depend on ground truth b) a reparametrization trick is used. I do not think, this justifies making the co-generative a central point of the paper. In my eyes, it would simply help the paper if these remarks were removed.

3) **Relevance for wider ICLR community** Given the very specialized domain, the work would perhaps be a better fit for a conference fully focused on computer vision. Note that e.g. (Fey, ICLR 2020) included also NLP experiments and made a more general point about matching procedures. I am afraid that in the current form, this paper won't attract too wide ICLR audiences. Is there a way to demonstrate/make-a-plausible-case-for wider applicability?

4) **At times unclear writeup** It wasn't always easy to understand what exactly the authors mean. Both in natural language and in technical parts. In the technical parts, the authors use very heavy notation and produce multiple large formulas that on the other have very standard content (e.g.. eq 11 and 14). On a technical level, the tools applied to introducing differentiability are very standard.  But that is absolutely fine since the core contribution is **where** they are applied.  There is no need to "produce enough formulas" such as by rewriting standard ELBO arguments over much larger notation (as it is in Appendix A3).

5) **Qualitative analysis** This is a missed opportunity. High-quality images that clearly demonstrate where the advantage is coming from, could easily be a highlight of this paper and should certianly be present in the main text. The current analysis in the supplementary is quite short and the displayed images are tiny, very hard to understand and **in almost all cases do NOT show the difference between the baseline and the proposed method.** I can't stress enough how instructive it would be to insert a section where eg. only examples of tables are analyzed (since that's the class with maximum gain) and one can clearly see what the architecture does differently (without zooming to 400%). Some parts of the technical presentation could be shortened.

Minor remarks follow:
- line 4 - footnote: It only is without loss of generality when negative costs are allowed, otherwise the argmax is always a maximal matching and outliers are not ignored.
- line 4 (LaTeX): This symbol for real numbers is very non-standard
- Figure 1 is hard to navigate. I suggest just focusing on the topology and dropping the matchings.
- page 3 - summation itself is undifferentiable - this is more misleading than anything. The point of this section is that one can introduce soft edges instead of the hard edges.
- page 4 - consistency loss - This needs an example or a figure for an explanation.
- page 4 - with full linearization (Swoboda, 2007) - This work doesn't linearize anything as far as I know. It is a SOTA combinatorial QAP solver based on message passing and dual ascent.
- Eq 11, I don't see any reason to include this. it is just a trivial factorization with heavy notation.
- Page 7, Figure 3 has poor visual quality and a very tiny font.

---

> ### Author Response · Authors · 2020-11-11
> **response to reviewer 5**
>
> We would like to thank the reviewer for his/her contributing suggestions and great effort on reviewing our paper. Our answers to the main concerns of the reviewer are as follows:
> 1. **Relevance to ICLR.** The reason that our experiments were conducted on computer vision tasks is because of the high quality and significant challenge of datasets in CV (e.g. Pascal VOC and SPair-71K). While the design of our method is not particularly for CV, we believe the main idea (of learning latent topology of structural data) is also helpful to several relevant tasks. In fact, several recent important papers on graph matching were also mainly focused on computer vision datasets [1,2] while have their counterparts in other non-CV problems (e.g. graph similarity [3] and Wassertein learning [4]).
> 2. **Qualitative analysis.** As suggested by the reviewer, we will add clearer and bigger figures in the updated version.
> 3. **Code.** We will release the source code once the paper can be accepted.
> 4. **Previous works.** Actually, we didn’t find any previous work for graph matching by estimating the latent topology as in our paper. We used “little” just to accomodate any possibility of such works. Since this can be misleading, we will correct it as “... there is no work on ...” instead. We will be happy to cite and discuss similar papers if reviewers can tell us any.
> 5. **Other concerns,** we will reorganize our paper and correct the typos to make it more friendly to readers. We will also avoid highlighting “co-generative” which might be misleading to the readers. We will also correct any portions that can be misleading.
>
> [1] Matthias Fey, Jan E Lenssen, Christopher Morris, Jonathan Masci, and Nils M Kriege. Deep graph matching consensus. In ICLR, 2020.
>
> [2] Tianshu Yu, Runzhong Wang, Junchi Yan, and Baoxin Li. Learning deep graph matching with channel-independent embedding and hungarian attention. In ICLR, 2020.
>
> [3] Yunsheng Bai, Hao Ding, Song Bian, Ting Chen, Yizhou Sun, and Wei Wang. SimGNN: A Neural Network Approach to Fast Graph Similarity Computation. WSDM, 2019.
>
> [4] Hongteng Xu, Dixin Luo, Hongyuan Zha, Lawrence Carin. Gromov-Wasserstein Learning for Graph Matching and Node Embedding. ICML, 2019.

---

> > ### Comment · AnonReviewer5 · 2020-11-11
> > **Quick response**
> >
> > I thank the authors for their quick response.
> >
> > I am convinced that the format of ICLR is a great opportunity to improve the review process for the whole community. One particular advantage of the rather long and interactive rebuttal period is that the authors can upload revisions of their paper. This eliminates the need to rely on "camera-ready promises" which are in many cases very unreliable (in particular with respect to code release).
> >
> > I do not see any reason why the authors cannot release code now as many other submissions already did -- it doesn't need to be in a github-ready state as far as I am concerned (and as is often the situation with other submissions). A similar argument goes for the reorganization of the paper, I will feel much more comfortable approving the updated version. Delivering it within the rebuttal period doesn't seem like an unreasonable request to me.
> >
> > In summary, I welcome that the authors reacted to my comments in such a cooperative fashion and I am only willing to improve the score but only after reviewing an updated manuscript.

---

### Author Response · Authors · 2020-11-11
**Thanks to all reviewers for their effort.**

Above all, we appreciate the significant effort of all the reviewers to deliver such detailed and constructive suggestions. We also appreciate that most reviewers identify the novelty of our work.

In particular, we see a main concern from several reviewers is about the organization of our work. Per the submission regulations, we can only upload an 8-page version of our manuscript for the initial submission, where some portion has to be put into the appendix. We would definitely reorganize the paper without greatly changing the content of the paper in the updated version allowing 9 pages.

Besides, some of the answers to the reviewers' concerns can be found in the paper, and we have stated the response to each reviewer separately as below. It would be so appreciated if reviewers can check our responses mutually.

We will take some time to update our manuscript and will get back to all reviewers once uploaded.

Again, thanks to all reviewers for their effort and time. We also welcome any suggestions and discussion from the third party.

---

### Author Response · Authors · 2020-11-14
**Revision uploaded and the change log is as follows**

We thank again all the reviewers' comments and effort.

We now have uploaded an updated version of the manuscript taking into account the reviewers' suggestions. While some of the questions to the reviewers' concerns can be found in the initial submission, we highlight them in the revision to make our paper friendly to the readers.

The change log of main updates is as follows.

### Introduction
We emphasize our **motivation** and **hypothesis** in the Sec 1 Introduction. We also give the definition of "topology" in our paper. Specifically, we add a footnote 2 to specify the loosely related work to ours, which are also discussed in detail in related works.

### Related works
We did not find any exact prior works that jointly *learn* latent topology and matching, while we do add two new references (Du et al 2019, 2020) which alternates link prediction and matching, and their method and problem setting are largely different from ours – see details in the paper.

### Paper organization
We move **Related works** before Sec 3 **Learning latent topology for GM** which may help the readers learn more background before diving into technical details.

### Figures
We move the holistic pipeline figure to the appendix A.1. Instead, we add a new Fig. 2(b) which illustrates how the consistency loss works and influences the matching, as suggested by the reviewer. Besides, we add much bigger figures in Table 7 of appendix A.5 showing how the generated topology differs from Delaunay.

### Minor changes
We correct the typos and some misleading expressions. We also add some explanations for the notations.

---

> ### Comment · AnonReviewer5 · 2020-11-19
> **Response to the new version**
>
> I appreciate the authors delivered an updated manuscript. In my eyes, it is a clear improvement over the previous version. Also, as I stated before, I am familiar with the benchmark and I find the achieved 2-3% margin non-trivial.
>
> There are two things that I still find unsatisfying:
>
> 1) The authors repeatedly refused to release code during the review period. To me, this is quite suspicious, particularly, since the implementation closely follows (Rolinek, 2020), whose implementation is publicly available. I find it hard to imagine that the learnable graph topology would expand this code by more than several hundred lines. Releasing this should be manageable during the review period, in my opinion.
>
> 2) I still don't understand what "really" makes the method work. Qualitatively, it seems to just occasionally add an edge to the Delauney triangulation... yet, for example on the class "Table", the difference in performance is drastic. Any attempt to dive into this and give a solid explanation would be of great interest to the community.

---

> > ### Author Response · Authors · 2020-11-19
> > **response**
> >
> > We thank reviewer5 for the response to our revised version of the manuscript. Upon the remaining concerns, our responses are as follows:
> >
> > 1. **Code.** We are now working on some new idea based on this submission, and trying to organize the code and release it within the decision period. However, at the current stage, we also wish reviewers can consider other factors (e.g. novelty, insight, motivation, theoretical soundness, improvement, etc.) when evaluating our paper.
> >
> > 2. **What really works.** In general, changing the topology not only affects the topology itself ($N_G$ in Fig. 2(a)), it also influences any other modules ($N_R$ and $N_B$ in Fig. 2(a)) since the framework is an end-to-end fashion (not step-by-step). In this sense, the learned node and edge features in both $N_R$ and $N_B$ can become more deterministic for matching task. The significant improvement on simple object "table" (where the topology generated from our module $N_G$ can be very similar to Delaunay) can be derived from such deterministic features by allowing new topology on complex object "human". So we infer that the latent topology can somewhat release some capacity of the whole network for complex object (e.g. human), and allow the released capacity to focus on objects with more ambiguity (e.g. table). The reason of this phenomenon is still unclear, and is related to a more essential question: **How the (latent) topology influences the feature learning in GNN backbones.** We will cover this topic in our future work. In summary, the latent topology is not produced occasionally, but **learned** by the network under some prior. Please also see ablation study in Appendix A.4, where we turn on different loss functions. We can suppose the performance from an occasionally created topology will not be affected by different loss functions, but we did observe each loss function can consistently improve the performance. This shows that the topology is not occasionally generated.
> >
> > Thank you again for the discussion and look forward to your reply.

---

### Decision · Program_Chairs · 2021-01-07
**Final Decision**

**Decision:**

Reject

**Comment:**

The paper proposes an interesting take on Graph Matching by posing the problem as learning the Topology through Graph Convolutional Networks.  There is consensus that the methods proposed are new but the impact is not clear.
One major point against the paper seems to be that Code is yet to be released.